# PROFEAT: PROJECTED FEATURE ADVERSARIAL TRAINING FOR SELF-SUPERVISED LEARNING OF ROBUST REPRESENTATIONS

## ABSTRACT

Supervised adversarial training has been the most successful approach for improving the robustness of Deep Neural Networks against adversarial attacks. While several recent works have attempted to overcome the need for supervision or labeled training data by integrating adversarial training with contrastive Self-Supervised Learning (SSL) approaches such as SimCLR, their performance has been sub-optimal due to the increased training complexity. A recent approach mitigates this by utilizing supervision from a standard self-supervised trained model in a teacher-student setting that mimics supervised adversarial training (Zhang et al., 2022). However, we find that there is still a large gap in performance when compared to supervised training, specifically on larger capacity models. We show that this is a result of mismatch in training objectives of the teacher and student, and propose Projected Feature Adversarial Training (ProFeAT) to bridge this gap by using a projection head in the adversarial training step. We further propose appropriate attack and defense losses at the feature and projector spaces, coupled with a combination of weak and strong augmentations for the teacher and student respectively, to improve generalization without increasing the training complexity. We demonstrate significant improvements in performance when compared to existing SSL methods, and performance on par with TRADES, a popular supervised adversarial training method, on several benchmark datasets and models.

## 1 INTRODUCTION

Deep Neural Networks are known to be vulnerable to crafted imperceptible input-space perturbations known as *Adversarial attacks* (Szegedy et al., 2013), which can be used to fool classification networks into predicting any desired output, leading to disastrous consequences. Amongst the diverse attempts at improving the adversarial robustness of Deep Networks, Adversarial Training (AT) (Madry et al., 2018; Zhang et al., 2019) has been the most successful. This involves the generation of adversarial attacks by maximizing the training loss, and further minimizing the loss on the generated attacks for training. While these methods have proved to be robust against various attacks developed over time (Carlini et al., 2019; Croce & Hein, 2020; Sriramanan et al., 2020), they require significantly more training data when compared to standard training (Schmidt et al., 2018), incurring a large annotation cost. Motivated by the success of contrastive learning for standard Self-Supervised Learning (SSL) (Van den Oord et al., 2018; Chen et al., 2020b; He et al., 2020), several works have attempted to use contrastive learning for self-supervised adversarial training as well (Jiang et al., 2020; Kim et al., 2020; Fan et al., 2021). While this strategy works well in a full network fine-tuning setting, the performance is sub-optimal when the robustly pretrained feature encoder is frozen while training the classification head (linear probing), demonstrating that the representations learned are indeed sub-optimal. A recent work, Decoupled Adversarial Contrastive Learning (DeACL) (Zhang et al., 2022), demonstrated significant improvements in performance and training efficiency by splitting this combined self-supervised adversarial training into two stages; first, where a standard self-supervised model is trained, and second, where this pretrained model is used as a teacher to provide supervision to the adversarially trained student network. Although the performance of this method is on par with supervised adversarial training on small model architectures (ResNet-18), we find that it does not scale to larger models such as WideResNet-34-10.

In this work, we aim to bridge the performance gap between self-supervised and supervised adversarial training methods, and improve the scalability of the former to larger model capacities. We utilize the distillation setting discussed above, where a standard self-supervised trained teacher provides supervision to the student. In contrast to a typical distillation scenario, the ideal goal for the student is not to replicate the teacher, but to leverage weak supervision from the teacher while simultaneously enhancing its adversarial robustness. This involves a trade-off between the sensitivity towards changes that flip the class of an image (for better clean accuracy) and invariance towards imperceptible perturbations that preserve the true class (for adversarial robustness) (Tramèr et al., 2020). Towards this, we propose to impose similarity with respect to the teacher in the appropriate dimensions by applying the distillation loss in a *projected space* (output of a projection MLP layer), while enforcing the smoothness-based robustness loss in the *feature space* (output of a backbone/feature extractor). However, we find that enforcing these losses at different layers results in training instability, and thus introduce the complementary loss (clean distillation loss or robustness loss) as a regularizer to improve training stability. We further propose to reuse the pretrained projection layer from the teacher model for better convergence.

In line with the training objective, the adversarial attack used during training aims to find images that maximize the smoothness loss in the feature space, and cause misalignment between the teacher and student in the projected space. Further, since data augmentations are known to increase the training complexity of adversarial training resulting in a drop in performance (Zhang et al., 2022; Addepalli et al., 2022), we propose to use augmentations such as AutoAugment (or *strong augmentations*) only at the student for better attack diversity, while using spatial transforms such as pad and crop (PC) (or *weak augmentations*) at the teacher. We summarize our contributions below:

- We propose Projected Feature Adversarial Training (ProFeAT) - a teacher-student distillation setting for self-supervised adversarial training, where the projection layer of the standard self-supervised pretrained teacher is reused for student distillation. We further propose appropriate attack and defense losses for training, coupled with a combination of weak and strong augmentations for the teacher and student respectively.

- Towards understanding *why* the projector helps, we first show that the compatibility between the training methodology of the teacher and the ideal goals of the student plays a crucial role in the student model performance in distillation. We further show that the use of a projector can alleviate the negative impact of the inherent misalignment of the above.

- We demonstrate the effectiveness of the proposed approach on the standard benchmark datasets CIFAR-10 and CIFAR-100. We obtain moderate gains on small model architectures (ResNet-18) and larger gains of $3.5 - 8\%$ in clean accuracy and $\sim 3\%$ in robust accuracy on larger models (WideResNet-34-10), while also outperforming TRADES supervised training (Zhang et al., 2019) on larger models.

## 2 PRELIMINARIES: PROBLEM SETTING AND NOTATION

We consider the problem of self-supervised learning of robust representations, where a self-supervised standard trained teacher model $\mathcal{T}$ is used to provide supervision to a student model $\mathcal{S}$. The feature, projector and linear probe layers of the teacher are denoted as $\mathcal{T}_f$, $\mathcal{T}_p$ and $\mathcal{T}_l$ respectively. An analogous notation is followed for the student as well. The dataset used for self-supervised pretraining $\mathcal{D}$ consists of images $x_i$ where $i \leq N$. An adversarial image corresponding to the image $x_i$ is denoted as $\tilde{x}_i$. We consider the $\ell_\infty$ based threat model where $||\tilde{x}_i - x_i||_\infty \leq \varepsilon$. The value of $\varepsilon$ is set to $8/255$ for CIFAR-10 and CIFAR-100 (Krizhevsky et al., 2009), as is standard in literature (Madry et al., 2018; Zhang et al., 2019). The Robust Accuracy (RA) in the SOTA comparison tables is presented against AutoAttack (Croce & Hein, 2020) (RA-AA) which is widely used as a benchmark for robustness evaluation (Croce et al., 2021). In all other tables, we present robust accuracy against the GAMA attack (Sriramanan et al., 2020) (RA-G) which is known to be competent with AutoAttack, while being significantly faster. We additionally present results against a 20 step PGD attack (Madry et al., 2018) (RA-PGD20), as is standard in self-supervised adversarial training literature (Fan et al., 2021; Zhang et al., 2022). We note that for comparing the robust accuracy between any two defenses, the accuracy against AutoAttack or GAMA should be considered. The accuracy gap between PGD-20 and Autoattack/ GAMA is higher when the loss surface is convoluted, due to the phenomenon of gradient masking Papernot et al. (2017); Tramèr et al. (2018). The accuracy on clean or natural samples is denoted as SA which stands for Standard Accuracy.

To evaluate the representations learned after self-supervised adversarial pretraining, we freeze the pretrained backbone, and perform linear layer training on a downstream labeled dataset consisting of image-label pairs. We refer to this training as linear probing (Kumar et al., 2022), as is common in literature. The training is done using cross-entropy loss on clean samples unless specified otherwise. We compare the robustness of the representations on both in-distribution data, where the linear probing is done using the same distribution of images as that used for pretraining, and in a transfer learning setting, where the distribution of images in the downstream dataset is different from that used for pretraining. We do not consider the case of fine-tuning the full network using adversarial training, since this changes the pretrained network to large extent, and may yield misleading results and conclusions depending on the dynamics of training (number of epochs, learning rate, and the value of the robustness-accuracy trade-off parameter). Contrary to this, linear probing based evaluation gives an accurate comparison of representations learned across different pretraining algorithms.

## 3    RELATED WORKS

**Self Supervised Learning (SSL):** With the abundance of unlabelled data, learning representations through self-supervision has seen major advances in recent years. Contrastive learning based SSL approaches have emerged as a promising direction (Van den Oord et al., 2018; Chen et al., 2020b; He et al., 2020), where different augmentations of a given anchor image form positives, and augmentations of other images in the batch form the negatives for training. The training objective involves pulling the representations of the positives together, and repelling the representations of negatives.

**Self Supervised Adversarial Training:** To alleviate the large sample complexity and training cost of adversarial training, there have been several works that have attempted self-supervised learning of adversarially robust representations (Kim et al., 2020; Jiang et al., 2020; Fan et al., 2021; Zhang et al., 2022). Chen et al. (2020a) propose AP-DPE, an ensemble adversarial pretraining framework where several pretext tasks like Jigsaw puzzles (Noroozi & Favaro, 2016), rotation prediction (Gidaris et al., 2018) and Selfie (Trinh et al., 2019) are combined to learn robust representations without task labels. Jiang et al. (2020) propose ACL, that combines the popular contrastive SSL method - SimCLR (Chen et al., 2020b) with adversarial training, using Dual Batch normalization layers for the student model - one for the standard branch and another for the adversarial branch. RoCL (Kim et al., 2020) follows a similar approach to ACL by combining the contrastive objective with adversarial training to learn robust representations. Fan et al. (2021) propose AdvCL, that uses high-frequency components in data as augmentations in contrastive learning, performs attacks on unaugmented images, and uses a pseudo label based loss for training to minimize the cross-task robustness transferability. Luo et al. (2023) study the role of augmentation strength in self-supervised contrastive adversarial training, and propose DynACL, that uses a "strong-to-weak" annealing schedule on augmentations. Additionally, motivated by Kumar et al. (2022), they propose DynACL++ that obtains pseudo-labels via k-means clustering on the clean branch of the DynACL pretrained network, and performs linear-probing (LP) using these pseudo-labels followed by adversarial full-finetuning (AFT) of the backbone. We note that the latter post-processing step is a generic finetuning strategy in literature that can be integrated with several base algorithms including ours.

While most self-supervised adversarial training methods aimed at integrating contrastive learning methods with adversarial training, Zhang et al. (2022) showed that combining the two is a very complex optimization problem due to their conflicting requirements. The authors propose Decoupled Adversarial Contrastive Learning (DeACL), where a teacher model is first trained using existing self-supervised training methods such as SimCLR, and further, a student model is trained to be adversarially robust using supervision from the teacher. While existing methods used $\sim 1000$ epochs for contrastive adversarial training, the compute requirement for DeACL is much lesser since the first contrastive learning stage does not involve adversarial training, and the second stage is similar in complexity to supervised adversarial training (Details in Appendix-F). We thus utilize this distillation framework and obtain significant gains over DeACL, specifically at larger model capacities.

## 4    PROPOSED METHOD

In this section, we motivate the need for a projection layer, and present the proposed approach.

### 4.1    PROJECTION LAYER IN SELF-SUPERVISED DISTILLATION

In this work, we follow the setting proposed by Zhang et al. (2022), where a standard self-supervised pretrained teacher provides supervision for self-supervised adversarial training of the student model.

Table 1: **Role of projector in self-supervised distillation (CIFAR-100, WRN-34-10):** The drop in accuracy of student ($\mathcal{S}$) w.r.t. the teacher ($\mathcal{T}$) indicates distillation performance, which improves by matching the training objective of the teacher with ideal goals of the student (S3/ S4 vs. S1), and by using similar losses for pretraining and linear probing (LP) (S2 vs. S1). Using a projector improves performance in case of mismatch in the above (S5 vs. S1). The similarity between teacher and student is significantly higher at the projector space when compared to the feature space in S5.

| Exp # | Teacher training | Teacher acc (%) | Projector | LP Loss | Student accuracy after linear probe | | $\cos(\mathcal{T}, \mathcal{S})$ | |
| | | | | | Feature space (%) | Projector space (%) | Feature space | Projector space |
|---|---|---|---|---|---|---|---|---|
| S1 | Self-supervised | 70.85 | Absent | CE | 64.90 | - | 0.94 | - |
| S2 | Self-supervised | 70.85 | Absent | $\cos(\mathcal{T}, \mathcal{S})$ | 68.49 | - | 0.94 | - |
| S3 | Supervised | 80.86 | Absent | CE | 80.40 | - | 0.94 | - |
| S4 | Supervised | 69.96 | Absent | CE | 71.73 | - | 0.98 | |
| S5 | Self-supervised | 70.85 | Present | CE | 73.14 | 64.67 | 0.19 | 0.92 |

This is different from a standard distillation setting (Hinton et al., 2015) because the representations of standard and adversarially trained models are known to be inherently different. Ilyas et al. (2019) attribute the adversarial vulnerability of models to the presence of non-robust features which can be disentangled from robust features that are learned by adversarially trained models. The differences in representations of standard and adversarially trained models can also be justified by the fact that linear probing of standard trained models using adversarial training cannot produce robust models as shown in Table-11. On a similar note, standard full finetuning of adversarially trained models destroys the robust features learned (Chen et al., 2020a; Kim et al., 2020; Fan et al., 2021), yielding $0\%$ robustness as shown in the table. Due to the inherently diverse representations of standard and adversarially trained models, the ideal goal of the student in the considered distillation setting is not to merely follow the teacher, but to be able to take weak supervision from it while being able to differ considerably. In order to achieve this, we take inspiration from standard self-supervised learning literature (Van den Oord et al., 2018; Chen et al., 2020b; He et al., 2020; Navaneet et al., 2022; Gao et al., 2022) and propose to utilize a projection layer following the student backbone, so as to isolate the impact of the enforced loss on the learned representations. Bordes et al. (2022) show that in standard supervised and self-supervised training, a projector is useful when there is a misalignment between the pretraining and downstream tasks, and aligning them can eliminate the need for the same. Motivated by this, we hypothesize the following for self-supervised distillation:

*Student model performance improves by matching the following during distillation:*

1. *Training objectives of the teacher and the ideal goals of the student,*
2. *Pretraining and linear probe training objectives of the student.*

The ideal goal of the student depends on the downstream task, which is standard accuracy in standard training, and standard and robust accuracy in adversarial training. The training objective of the teacher is to achieve invariance to augmentations of the same image when compared to augmentations of other images in contrastive SSL training, and standard accuracy in supervised training. We explain the intuition behing the hypotheses in Appendix-B, and empirically justify the same by considering several distillation settings involving standard and adversarial, supervised and self-supervised trained teacher models in Tables-1 and 2. The results are presented on CIFAR-100 (Krizhevsky et al., 2009) with WideResNet-34-10 (Zagoruyko & Komodakis, 2016) architecture for both teacher and student. The standard self-supervised model is trained using SimCLR (Chen et al., 2020b). Contrary to a typical knowledge distillation setting where a cross-entropy loss is also used (Hinton et al., 2015), all the experiments presented involve the use of only self-supervised losses for distillation (cosine similarity between representations), and labels are used only during linear probing. Adversarial self-supervised distillation in Table-2 is performed using a combination of distillation loss on natural samples and smoothness loss on adversarial samples as shown in Eq.2 (Zhang et al., 2022). A randomly initialized trainable projector is used at the output of student backbone in S5 of Table-1 and A4 of Table-2. Here, the training loss is considered in the projected space of the student ($\mathcal{S}_p$) rather than the feature space ($\mathcal{S}_f$).

**1. Matching the training objectives of teacher with the ideal goals of the student:** We first consider the standard training of a student model, using either a self-supervised or supervised teacher in Table-1. In the absence of a projector, the drop in student accuracy w.r.t. the respective teacher accuracy is $6\%$ with a self-supervised teacher (S1), and $< 0.5\%$ with a supervised teacher (S3). To ensure that our observations are not a result of the $10\%$ difference in teacher accuracy between S1 and S3, we present results and similar observations with a supervised sub-optimally trained teacher

Table 2: **Role of projector in self-supervised adversarial distillation (CIFAR-100, WRN-34-10):** Student performance after linear probe at feature space is reported. The drop in standard accuracy (SA) of the student ($\mathcal{S}$) w.r.t. the teacher ($\mathcal{T}$), and the robust accuracy (RA-G) of the student improve by matching the training objective of the teacher with ideal goals of the student (A3 vs. A1), and by using similar losses for pretraining and linear probing (LP) (A2 vs. A1). Using a projector improves performance in case of mismatch in the above (A4 vs. A1).

| Exp # | Teacher training | Teacher accuracy | | Projector | LP Loss | Student accuracy | | $\cos(\mathcal{T}, \mathcal{S})$ |
|---|---|---|---|---|---|---|---|---|
| | | SA (%) | RA-G (%) | | | SA (%) | RA-G (%) | |
| A1 | Self-supervised (standard training) | 70.85 | 0 | Absent | CE | 50.71 | 24.63 | 0.78 |
| A2 | Self-supervised (standard training) | 70.85 | 0 | Absent | $\cos(\mathcal{T}, \mathcal{S})$ | 54.48 | 23.20 | 0.78 |
| A3 | Supervised (TRADES adversarial training) | 59.88 | 25.89 | Absent | CE | 54.86 | 27.17 | 0.94 |
| A4 | Self-supervised (standard training) | 70.85 | 0 | Present | CE | 57.51 | 24.10 | 0.18 |

in S4. Thus, a supervised teacher is significantly better than a self-supervised teacher for distilling representations specific to a given task. This justifies the hypothesis that, *student performance improves by matching the training objectives of the teacher and the ideal goals of the student*.

We next consider adversarial training of a student, using either a standard self-supervised teacher, or a supervised adversarially trained teacher (TRADES (Zhang et al., 2019)) in Table-2. Since the TRADES model is more aligned with the ideal goals of the student, despite its sub-optimal clean accuracy, the clean and robust accuracy of the student are better than those obtained using a standard self-supervised model as a teacher (A3 vs. A1). This further justifies the first hypothesis.

**2. Matching the pretraining and linear probe training objectives of the student:** To align pretraining with linear probing, we perform linear probing on the teacher model, and further train the student by maximizing the cosine similarity between the logits of the teacher and student. This boosts the student accuracy by 3.6%, in Table-1 (S2 vs. S1) and by $3.8\%$ in Table-2 (A2 vs. A1).

The projector isolates the representations of the student from the training loss, as indicated by the lower similarity between the student and teacher at feature space when compared to that at the projector (in S5 and A4), and prevents overfitting of the student to the teacher training objective. This makes the student robust to the misalignment between the teacher training objective and ideal goals of the student, and also to the mismatch in student pretraining and linear probing objectives, thereby improving student performance, as seen in Tables-1 (S5 vs. S1) and 2 (A4 vs. A1).

## 4.2 PROFEAT: PROJECTED FEATURE ADVERSARIAL TRAINING

We present details on the proposed approach Projected Feature Adversarial Training, illustrated in Fig.1. Firstly, a teacher model is trained using a self-supervised training algorithm such as SimCLR (Chen et al., 2020b), which is also used as an initialization for the student for better convergence.

**Use of Projection Layer:** As discussed in Section-4.1, to overcome the impact of the inherent misalignment between the training objective of the teacher and the ideal goals of the student, and the mismatch between the pretraining and linear probing objectives, we propose to use a projection head at the output of the student backbone. As noted in Tables-1 (S5 vs. S1) and 2 (A4 vs. A1), even a randomly initialized projection head improves performance. Most self-supervised pretraining methods use similarity based losses at the output of a projection head for training (Chen et al., 2020b; He et al., 2020; Grill et al., 2020; Chen & He, 2021; Zbontar et al., 2021), resulting in a projected space where similarity has been enforced during pretraining, thus giving higher importance to the key dimensions. We therefore propose to reuse this pretrained projection head for both teacher and student and freeze it during training to prevent convergence to an identity mapping.

**Defense loss:** As is common in adversarial training literature (Zhang et al., 2019; 2022), we use a combination of loss on clean samples and smoothness loss to enforce adversarial robustness in the student model. Since the loss on clean samples utilizes supervision from the self-supervised pretrained teacher, it is enforced at the outputs of the respective projectors of the teacher and student as discussed above. The goal of the second loss is merely to enforce local smoothness in the loss surface of the student, and is enforced in an unsupervised manner (Zhang et al., 2019; 2022). Thus, it is ideal to enforce this loss at the feature space of the student network, since these representations are directly used for downstream applications. While the ideal locations for the clean and adversarial losses are the projected and feature spaces respectively, we find that such a loss formulation is hard to optimize, resulting in either a non-robust model, or collapsed representations as shown in Table-

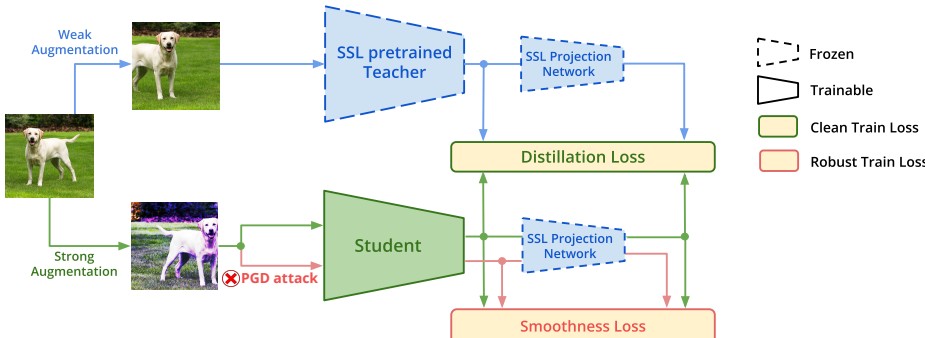

Figure 1: **Proposed approach (ProFeAT):** The student is trained using a distillation loss on clean samples using supervision from an SSL pretrained teacher, and a smoothness loss to enforce adversarial robustness. A frozen pretrained projection layer is used at the teacher and student to prevent overfitting to the clean distillation loss. The use of strong augmentations at the student increases attack diversity, while weak augmentations at the teacher reduce the training complexity.

12. We therefore utilize a complimentary loss as a regularizer in the respective spaces. This results in a combination of losses at the feature and projector spaces as shown below:

$$\mathcal{L}_{fp} = -\sum_i \cos\left(\mathcal{T}_{fp}(x_i), \mathcal{S}_{fp}(x_i)\right) + \beta \cdot \cos\left(\mathcal{S}_{fp}(x_i), \mathcal{S}_{fp}(\tilde{x}_i)\right) \tag{1}$$

$$\mathcal{L}_f = -\sum_i \cos\left(\mathcal{T}_f(x_i), \mathcal{S}_f(x_i)\right) + \beta \cdot \cos\left(\mathcal{S}_f(x_i), \mathcal{S}_f(\tilde{x}_i)\right) \tag{2}$$

$$\mathcal{L}_{\text{ProFeAT}} = \frac{1}{2} \cdot \left(\mathcal{L}_{fp} + \mathcal{L}_f\right) \tag{3}$$

$$\tilde{x}_i = \underset{\tilde{x}_i : ||\tilde{x}_i - x_i||_\infty \leq \varepsilon}{\arg\max} -\left\{ \cos\left(\mathcal{T}_{fp}(x_i), \mathcal{S}_{fp}(\tilde{x}_i)\right) + \cos\left(\mathcal{S}_f(x_i), \mathcal{S}_f(\tilde{x}_i)\right) \right\} \tag{4}$$

Here, $\mathcal{T}_{fp}$ is a composition of the feature backbone $\mathcal{T}_f$ and the projection layer $\mathcal{T}_p$ of the teacher, and a similar notation is used for the student as well. The first term in Eq.1 and 2 represents the clean loss, which is the cosine similarity between the representations of the teacher and the student at corresponding layers. The second term corresponds to the smoothness loss at the respective layers of the student, and is weighted by a hyperparameter $\beta$ that controls the robustness-accuracy trade-off in the downstream model. The overall loss $\mathcal{L}_{\text{ProFeAT}}$ is an equally weighted combination of losses at the feature and projection spaces as shown in Eq.3, and is minimized during training. We show in Fig.4 that the model is stable to variations in weighting between the losses. We conjecture that the pretrained projection layer steers the cosine similarity loss to give higher importance to the required features, although similarity is additionally enforced on the feature layer as well.

**Attack generation:** The attack used in the training loss is generated by a maximizing a combination of losses in both projector and feature spaces as shown in Eq.4. Since the projector space is primarily used for enforcing similarity with the teacher, we minimize the cosine similarity between the teacher and student representations for attack generation. Since the feature space is primarily used for enforcing local smoothness in the loss surface of the student, we utilize the unsupervised formulation that minimizes similarity between representations of clean and adversarial samples at the student.

**Augmentations:** Standard supervised and self-supervised training approaches are known to benefit from the use of strong data augmentations such as AutoAugment (Cubuk et al., 2018). However, such augmentations, which distort the low-level features of images, are known to deteriorate the performance of adversarial training (Rice et al., 2020; Gowal et al., 2020). Addepalli et al. (2022) attribute the poor performance to the larger domain shift between the augmented train and unaugmented test set images, in addition to the increased complexity of the adversarial training task, which overpower the superior generalization attained due to the use of diverse augmentations. Although these factors influence adversarial training in the self-supervised regime as well, we hypothesize that the need for better generalization is higher in self-supervised training, since the pretraining task is not aligned with the ideal goals of the student, making it important to use strong augmentations.

Table 3: **SOTA comparison:** Standard Linear Probing performance (%) on CIFAR-10 and CIFAR-100 datasets on ResNet-18 and WideResNet-34-10 models. Mean and standard deviation across 3 reruns are reported for DeACL Zhang et al. (2022) and the proposed approach, ProFeAT. Standard Accuracy (SA), Robust Accuracy against AutoAttack (RA-AA) and PGD-20 (RA-PGD20) reported.

| | CIFAR-10 | | | CIFAR-100 | | |
|---|---|---|---|---|---|---|
| | **SA** | **RA-PGD20** | **RA-AA** | **SA** | **RA-PGD20** | **RA-AA** |
| | | | **ResNet-18** | | | |
| Supervised (TRADES) | 83.74 | 49.35 | 47.60 | 59.07 | 26.22 | 23.14 |
| AP-DPE | 78.30 | 18.22 | 16.07 | 47.91 | 6.23 | 4.17 |
| RoCL | 79.90 | 39.54 | 23.38 | 49.53 | 18.79 | 8.66 |
| ACL | 77.88 | 42.87 | 39.13 | 47.51 | 20.97 | 16.33 |
| AdvCL | 80.85 | 50.45 | 42.57 | 48.34 | 27.67 | 19.78 |
| DynACL | 77.41 | - | 45.04 | 45.73 | - | 19.25 |
| DynACL++ | 79.81 | - | 46.46 | 52.26 | - | 20.05 |
| DeACL (Reported) | 80.17 | 53.95 | 45.31 | 52.79 | 30.74 | 20.34 |
| DeACL (Our Teacher) | $80.05_{\pm 0.29}$ | $52.97_{\pm 0.08}$ | $\mathbf{48.15}_{\pm 0.05}$ | $51.53_{\pm 0.30}$ | $30.92_{\pm 0.21}$ | $21.91_{\pm 0.13}$ |
| ProFeAT (**Ours**) | $\mathbf{81.68}_{\pm 0.23}$ | $49.55_{\pm 0.16}$ | $47.02_{\pm 0.01}$ | $\mathbf{53.47}_{\pm 0.10}$ | $27.95_{\pm 0.13}$ | $\mathbf{22.61}_{\pm 0.14}$ |
| | | | **WideResNet-34-10** | | | |
| Supervised (TRADES) | 85.50 | 54.29 | 51.59 | 59.87 | 28.86 | 25.72 |
| DynACL++ (500 epochs) | 82.27 | 49.60 | 47.12 | 52.59 | 24.22 | 21.27 |
| DynACL++ (1000 epochs) | 80.97 | 48.28 | 45.50 | 52.60 | 23.42 | 20.58 |
| DeACL | $83.83_{\pm 0.20}$ | $57.09_{\pm 0.06}$ | $48.85_{\pm 0.11}$ | $52.92_{\pm 0.35}$ | $32.66_{\pm 0.08}$ | $23.82_{\pm 0.07}$ |
| ProFeAT (**Ours**) | $\mathbf{87.62}_{\pm 0.13}$ | $54.50_{\pm 0.17}$ | $\mathbf{51.95}_{\pm 0.19}$ | $\mathbf{61.08}_{\pm 0.18}$ | $31.96_{\pm 0.08}$ | $\mathbf{26.81}_{\pm 0.11}$ |

Table 4: **Performance across different models:** Standard Linear Probing performance (%) of DeACL (Baseline) and ProFeAT (Ours) across different architectures on CIFAR-100. ViT-B/16 uses Imagenet-1K trained SSL teacher for training, while the SSL teacher in all other cases is trained on the CIFAR-100. SA: Standard Accuracy, RA-AA: Robust Accuracy against AutoAttack.

| | # parameters (M) | DeACL | | ProFeAT (Ours) | |
|---|---|---|---|---|---|
| | | **SA** | **RA-AA** | **SA** | **RA-AA** |
| ResNet-18 | 11.27 | 51.53 | 21.91 | **53.47** | **22.61** |
| ResNet-50 | 23.50 | 53.30 | 23.00 | **59.34** | **25.86** |
| WideResNet-34-10 | 46.28 | 52.92 | 23.82 | **61.08** | **26.81** |
| ViT-B/16 | 85.79 | 61.34 | 17.49 | **65.08** | **21.52** |

However, it is also important to ensure that the training task is not too complex. We thus propose to use a combination of weak and strong augmentations as inputs to the teacher and student respectively, as shown in Fig.1. From Fig.2 we note that, the use of strong augmentations results in the generation of more diverse attacks, resulting in a larger drop when differently augmented images are used across different restarts of a PGD 5-step attack. The use of weak augmentations at the teacher imparts better supervision to the student, reducing the training complexity.

# 5 EXPERIMENTS AND RESULTS

## 5.1 COMPARISON WITH THE STATE-OF-THE-ART

In Table-3, we present a comparison of the proposed approach ProFeAT with respect to several existing self-supervised adversarial training approaches (Chen et al., 2020a; Kim et al., 2020; Jiang et al., 2020; Fan et al., 2021; Zhang et al., 2022; 2019) by freezing the feature extractor and performing linear probing using cross-entropy loss on clean samples. To ensure a fair comparison, the same is done for the supervised AT method TRADES (Zhang et al., 2019) as well. We report results on CIFAR-10 and CIFAR-100 datasets, and on ResNet-18 and WideResNet-34-10 architectures. The results of existing methods on ResNet-18 architecture are as reported by Zhang et al. (2022). Since DeACL (Zhang et al., 2022) also uses a teacher-student architecture, we reproduce their results using the same teacher as our method, and report the same as "DeACL (Our Teacher)". Since most existing methods do not report results on WideResNet-34-10, we compare our results only with the best performing method (DeACL) and a recent method DynACL Luo et al. (2023). These results are not reported in the respective papers, hence we run them using the official code.

The proposed approach obtains competent robustness-acurracy trade-off when compared to the best performing baseline method DeACL on CIFAR-10 dataset with ResNet-18 architecture, and obtains $\sim 2\%$ higher clean accuracy alongside marginal gains in robust accuracy on CIFAR-100 ResNet-18.

Table 5: **Transfer Learning:** Standard Linear Probing performance (%) for transfer learning from CIFAR-10 and CIFAR-100 to STL-10 dataset on ResNet-18 and WideResNet-34-10 models. Standard Accuracy (SA), robustness against PGD-20 (RA-PGD20) and AutoAttack (RA-AA) reported.

| | ResNet-18 | | | | | | | WideResNet-34-10 | | | | | |
| | CIFAR-10 ->STL-10 | | | CIFAR-100 ->STL-10 | | | | CIFAR-10 ->STL-10 | | | CIFAR-100 ->STL-10 | | |
| | SA | RA-PGD20 | RA-AA | SA | RA-PGD20 | RA-AA | | SA | RA-PGD20 | RA-AA | SA | RA-PGD20 | RA-AA |
|---|---|---|---|---|---|---|---|---|---|---|---|---|---|
| Supervised | 54.70 | 30.45 | 22.26 | 51.11 | 23.63 | 19.54 | Supervised | 67.15 | 32.78 | 30.49 | **57.68** | 17.49 | 11.26 |
| DeACL | 60.10 | 41.40 | 30.71 | 50.91 | 27.76 | 16.25 | DeACL | 66.45 | 39.28 | 28.43 | 50.59 | 27.50 | 13.49 |
| ProFeat (Ours) | **64.30** | 35.50 | **30.95** | **52.63** | 26.72 | **20.55** | ProFeat (Ours) | **69.88** | 35.48 | **31.65** | 56.68 | 24.95 | **19.46** |

Table 6: **Ablations on Projector (CIFAR-100, WRN-34-10):** Performance (%) using variations in projector (proj.) initialization (init.) and trainability. SA: Standard Accuracy, RA-G: Robust accuracy against GAMA, RA-PGD20: Robust Accuracy against PGD-20 attack

| # | Student proj. | Proj. init. (student) | Teacher proj | Proj. init. (teacher) | SA | RA-PGD20 | RA-G |
|---|---|---|---|---|---|---|---|
| AP1 | Absent | - | Absent | - | 55.35 | 35.89 | **27.86** |
| AP2 | Trainable | Random | Absent | - | **63.07** | 32.05 | 26.57 |
| AP3 | Frozen | Pretrained | Absent | - | 40.43 | 27.51 | 22.23 |
| AP4 | Trainable | Pretrained | Absent | - | 62.89 | 31.97 | 26.57 |
| AP5 | Trainable | Random (common) | Trainable | Random (common) | 53.43 | 35.58 | 27.23 |
| **Ours** | Frozen | Pretrained | Frozen | Pretrained | 61.05 | 31.99 | 27.41 |
| AP6 | Trainable | Pretrained (common) | Trainable | Pretrained (common) | 54.60 | 36.10 | 27.41 |
| AP7 | Trainable | Pretrained | Frozen | Pretrained | 58.18 | 35.26 | 27.73 |

On WideResNet-34-10 architecture, we obtain $\sim 3 - 3.5\%$ gains in both robust and clean accuracy on CIFAR-10, and similar gains in robust accuracy of CIFAR-100 as well. We obtain exceptional gains of $\sim 8\%$ on the clean accuracy of CIFAR-100. Overall, the gains of the proposed approach are higher for larger model capacities (WRN-34-10). We obtain superior results when compared to the supervised AT method TRADES as well, at higher model capacities. We present an evaluation of the pretrained models using other methods such as KNN in Appendix-G.8.

**Results of CIFAR-100 across different model architectures:** We report performance of the proposed method ProFeAT and the best baseline DeACL on diverse architectures including Vision transformers Dosovitskiy et al. (2021) on the CIFAR-100 dataset in Table-4. ProFeAT consistently outperforms DeACL in both clean and robust accuracy across various model architectures. (See Appendix-C for details)

**Transfer learning:** To evaluate the robustness and generalization of the representations learned, we compare the proposed approach with the best baseline DeACL (Zhang et al., 2022) in Table-5. We consider transfer from CIFAR-10 to STL-10 (Coates et al., 2011) and CIFAR-100 to STL-10. When compared to DeACL, the clean accuracy is $\sim 4 - 10\%$ higher on CIFAR-10 and $\sim 1.7 - 6\%$ higher on CIFAR-100. We also obtain $3 - 5\%$ higher robust accuracy when compared to DeACL on CIFAR-100, and higher improvements over TRADES. Results with Adversarial Full Finetuning (AFF) to STL-10 and Caltech-101 are presented in Appendix-G.9.

## 5.2 ABLATIONS

**Projection Layer:** We present ablation experiments using different configurations of the projection layer in Table-6. As discussed in Section-4.1, we observe a large boost in clean accuracy when a random (or pretrained) trainable projection layer is introduced to the student (AP2/ AP4 vs. AP1). While the use of pretrained frozen projection head only for the student degrades performance considerably (AP3), the use of the same for both teacher and student (Ours) yields a optimal robustness-accuracy trade-off across all variations. The use of a common trainable projection head for both teacher and student results in collapsed representations at the projector output (AP5, AP6), yielding results similar to the case where projector is not used for both teacher and student (AP1). This issue is overcome when the pretrained projector is trainable only for the student (AP7).

**Training Loss:** We present ablation experiments across variations in training loss at the feature space and the projection head in Table-7. In the proposed approach (Ours), we introduce a combination of clean and robust losses at both feature and projector layers, as shown in Eq.3. By introducing the loss only at the features (AD1), there is a considerable drop in clean accuracy as seen earlier, which can be recovered by introducing the clean loss at the projection layer (AD3). Instead, when only the robust loss is introduced at the projection layer (AD4), there is a large drop in clean accuracy confirming that the need for projection layer is mainly enforcing the clean loss. When the combined loss is enforced only at the projection head (AD2), the accuracy is close to that of the

Table 7: **Ablations on Training Loss (CIFAR-100, WRN-34-10):** Performance (%) with variations in training loss at feature (feat.) and projector (proj.). "Clean" denotes the cosine similarity between representations of teacher and student on clean samples. "Adv" denotes the cosine similarity between representations of the corresponding clean and adversarial samples either at the output of student $(\mathcal{S}, \mathcal{S})$ or between the teacher and student $(\mathcal{T}, \mathcal{S})$. SA: Standard Accuracy, RA-G: Robust accuracy against GAMA, RA-PGD20: Robust Accuracy against PGD-20 attack

| # | Loss @ feat. | Loss @ proj. | SA | RA-PGD20 | RA-G | # | Defense @ feat | Defense @ proj | SA | RA-PGD20 | RA-G |
|---|---|---|---|---|---|---|---|---|---|---|---|
| **Ours** | clean + adv$(\mathcal{S},\mathcal{S})$ | clean + adv$(\mathcal{S},\mathcal{S})$ | 61.05 | 31.99 | 27.41 | AD5 | adv$(\mathcal{S},\mathcal{S})$ | clean | 59.72 | 3.77 | 1.38 |
| AD1 | clean + adv$(\mathcal{S},\mathcal{S})$ | - | 55.35 | 35.89 | **27.86** | AD6 | adv$(\mathcal{S},\mathcal{S})$ | clean + adv$(\mathcal{S},\mathcal{S})$ | 59.22 | 34.08 | 26.50 |
| AD2 | - | clean + adv$(\mathcal{S},\mathcal{S})$ | 59.65 | 33.03 | 26.90 | AD7 | clean | clean + adv$(\mathcal{S},\mathcal{S})$ | 62.24 | 30.55 | 25.97 |
| AD3 | clean + adv$(\mathcal{S},\mathcal{S})$ | clean | 61.69 | 31.34 | 26.40 | AD8 | clean + adv$(\mathcal{T},\mathcal{S})$ | clean + adv$(\mathcal{T},\mathcal{S})$ | 63.85 | 29.97 | 23.91 |
| AD4 | clean + adv$(\mathcal{S},\mathcal{S})$ | adv$(\mathcal{S},\mathcal{S})$ | 49.59 | 31.79 | 25.35 | AD9 | clean + adv$(\mathcal{T},\mathcal{S})$ | clean + adv$(\mathcal{T},\mathcal{S})$ | **65.34** | 27.75 | 22.40 |

Table 8: **Ablations on Augmentations used (CIFAR-100, WRN-34-10):** Performance (%) using different augmentations for the teacher and student. (PC: Pad+Crop, AuAu: AutoAugment). Standard Accuracy (SA) and Robust accuracy against GAMA (RA-G), PGD-20 (RA-PGD20) reported

| # | Teacher | Student | SA | RA-PGD20 | RA-G | # | Teacher | Student | SA | RA-PGD20 | RA-G |
|---|---|---|---|---|---|---|---|---|---|---|---|
| AG1 | PC | PC | 56.57 | 30.54 | 25.29 | AG4 | AuAu1 | AuAu2 | 59.51 | 32.44 | **28.15** |
| AG2 | AuAu | AuAu | 60.76 | 31.83 | 27.21 | AG5 | AuAu | PC | 57.28 | 31.23 | 26.14 |
| AG3 | PC1 | PC2 | 56.95 | 30.94 | 25.39 | **Ours** | PC | AuAu | **61.05** | 31.99 | 27.41 |

proposed approach, with marginally lower clean and robust accuracy. Enforcing only adversarial loss in the feature space, and only clean loss in the projector space is a hard optimization problem, and this results in a non-robust model (AD5). As shown in Table-12, even by increasing $\beta$ in AD5, we do not obtain a robust model, rather, there is a representation collapse. Thus, as discussed in Section-4.1, it is important to introduce the adversarial loss as a regularizer in the projector space as well (AD6). Enforcing only one of the two losses at the feature space (AD6 and AD7) also results in either inferior clean accuracy or robustness. Finally from AD8 and AD9 we note that the robustness loss is better when implemented as a smoothness constraint on the representations of the student, rather than by matching representations between the teacher and student. Overall, the proposed approach (Ours) results in the best robustness-accuracy trade-off.

**Training Augmentations:** We present ablation experiments to understand the impact of augmentations used for the teacher and student in Table-8. The base method (AG1) uses common Pad and Crop (PC) augmentation for both teacher and student. By using more complex augmentations -AutoAugment followed by Pad and Crop (denoted as AuAu in the table), there is a significant improvement in both clean and robust accuracy. By using separate augmentations for the teacher and student, there is an improvement in the case of PC (AG3), but a drop in clean accuracy accompanied by better robustness in case of AuAu. Finally by using a mix of both AuAu and PC at the student and teacher respectively (Ours), we obtain improvements in both clean and robust accuracy, since the former improves attack diversity as shown in Fig.2, while the latter makes the training task easier.

**Robustness-Accuracy trade-off:** We present results across variation in the robustness-accuracy trade-off parameter $\beta$ (Eq.1 and 2) in Fig.3. Both robustness and accuracy of the proposed method are significantly better than DeACL across all values of $\beta$. Secondly, the proposed approach allows a significantly better control over the robustness-accuracy trade-off, specifically in the range 2-12, where the linear drop in clean accuracy is accompanied by an increase in the robust accuracy.

We present additional ablations by varying the SSL algorithm, projection layer architecture, number of training epochs of the teacher, and number of attack steps in Appendix-G.

## 6 CONCLUSION

In this work, we bridge the performance gap between supervised and self-supervised adversarial training approaches, specifically for large capacity models. We utilize a teacher-student setting (Zhang et al., 2022) where a standard self-supervised trained teacher is used to provide supervision to the student. Due to the inherent misalignment between the teacher training objective and the ideal goals of the student, we propose to use a projection layer in order to prevent the network from overfitting to the standard SSL trained teacher. We present a detailed analysis on the use of projection layer in distillation to justify our method. We additionally propose appropriate attack and defense losses in the feature and projector spaces alongside the use of weak and strong augmentations for the teacher and student respectively, to improve the attack diversity while maintaining low complexity of the training task. The proposed approach obtains significant gains over existing self-supervised adversarial training methods, specifically at large model capacities, demonstrating its scalability.

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

## A  BACKGROUND: SUPERVISED ADVERSARIAL DEFENSES

Following the demonstration of adversarial attacks by Szegedy et al. (2013), there have been several attempts of defending Deep Networks against them. Early defenses proposed intuitive methods that introduced non-differentiable or randomized components in the network to thwart gradient-based attacks (Buckman et al., 2018; Ma et al., 2018; Dhillon et al., 2018; Xie et al., 2018; Song et al., 2018). While these methods were efficient and easy to implement, Athalye et al. (2018) proposed adaptive attacks which successfully broke several such defenses by replacing the non-differentiable components with smooth differentiable approximations, and by taking an expectation over the randomized components. Adversarial Training (Madry et al., 2018; Zhang et al., 2019) was the most successful defense strategy that withstood strong white-box (Croce & Hein, 2020; Sriramanan et al., 2020), black-box (Andriushchenko et al., 2020; Chen & Gu, 2020) and adaptive attacks (Athalye et al., 2018; Tramer et al., 2020) proposed over the years. PGD (Projected Gradient Descent) based adversarial training (Madry et al., 2018) involves maximizing the cross-entropy loss to generate adversarial attacks, and further minimizing the loss on the adversarial attacks for training. Another successful supervised Adversarial Training based defense was TRADES (Zhang et al., 2019), where the Kullback-Leibler (KL) divergence between clean and adversarial samples was minimized along with the cross-entropy loss on clean samples for training. Adjusting the weight between the losses gave a flexible trade-off between the clean and robust accuracy of the trained model. Although these methods have been robust against several attacks, it has been shown that the sample complexity of adversarial training is large (Schmidt et al., 2018), and this increases the training and annotation costs needed for adversarial training.

## B  DETAILS ON THE HYPOTHESES

In this section, we justify the intuition behind the hypotheses presented in Section-4.1, which is restated below:

*Student model performance improves by matching the following during distillation:*

1. *Training objectives of the teacher and the ideal goals of the student,*
2. *Pretraining and linear probe training objectives of the student.*

**Hypothesis-1:** Consider task-A to be the teacher's training task, and task-B to be the student's downstream task or its ideal goal. The representations in deeper layers (last few layers) of the teacher are more tuned to its training objective, and the early layers contain a lot more information than what is needed for this task Bordes et al. (2022). Thus, features specific to task-A are dominant or replicated in the final feature layer, and other features that may be relevant to task-B are sparse. When a similarity based distillation loss is enforced on such features, higher importance is given to matching the replicated features, and the sparse features which may be important for task-B are suppressed further in the student Addepalli et al. (2023). On the other hand, when the student's task matches with the teacher's task, a similarity based distillation loss is very effective in transferring the necessary representations to the student, since they are predominant. Thus, matching the training objective of the teacher with the ideal goals of the student improves downstream performance.

**Hypothesis-2:** For a given network, aligning the pretraining task with downstream task results in better performance since the matching of tasks ensures that the required features are predominant, and they are easily used by an SVM classifier (or a linear classifier) trained over it Addepalli et al. (2023). In context of distillation, since the features of the student are trained by enforcing similarity based loss w.r.t. the teacher, we hypothesize that enforcing similarity w.r.t. the teacher is the best

way to learn the student classifier as well. To illustrate this, we consider task-A to be the teacher pretraining task, and task-B to be the downstream task or ideal goal of the student. As discussed above, the teacher's features are aligned to task-A and these are transferred effectively to the student. The features related to task-B are suppressed in the teacher and are further suppressed in the student. As the features specific to a given task become more sparse, it is harder for an SVM classifier (or a linear classifier) to rely on that feature, although it important for classification Addepalli et al. (2023). Thus, training a linear classifier for task-B is more effective on the teacher when compared to the student. The linear classifier of the teacher in effect amplifies the sparse features, allowing the student to learn them more effectively. Thus, training a classifier on the teacher and distilling it to the student is better than training a classifier directly on the student.

We empirically justify our hypotheses in Section-4.1.

## C  MECHANISM FOR SCALING TO LARGER DATASETS

For a sufficiently complex task, a scalable approach results in better performance on larger models given enough data. Although the task complexity of adversarial self-supervised learning is high, the gains in prior approaches are marginal with an increase in model size, while the proposed method results in significantly improved performance on larger capacity models (Table-3). We discuss the key factors that result in better scalability below:

- As discussed in Section-4.1, a mismatch between training objectives of the teacher and ideal goals of the student causes a drop in student performance. This primarily happens because of the overfitting to the teacher training task. As model size increases, the extent of overfitting increases. The use of a projection layer during distillation alleviates the impact of this overfitting and allows the student to retain more generic features that are useful for the downstream robust classification objective. Thus, a projection layer is more important for larger model capacities where the extent of overfitting is higher.

- Secondly, as the model size increases, there is a need for higher amount of training data for achieving better generalization. The proposed method has better data diversity as it enables the use of more complex data augmentations in adversarial training by leveraging supervision from weak augmentations at the teacher.

## D  DETAILS ON DATASETS

We compare the performance of the proposed approach ProFeAT with existing methods on the benchmark datasets CIFAR-10 and CIFAR-100 Krizhevsky et al. (2009), that are commonly used for evaluating the adversarial robustness of models Croce et al. (2021). Both datasets consist of RGB images of dimension $32 \times 32$. CIFAR-10 consists of 50,000 images in the training set and 10,000 images in the test set, with the images being divided equally into 10 classes - airplane, automobile, bird, cat, deer, dog, frog, horse, ship and truck. CIFAR-100 dataset is of the same size as CIFAR-10, with images being divided equally into 100 classes. Due to the larger number of classes, there are only 500 images per class in CIFAR-100, making it a more challenging dataset when compared to CIFAR-10.

## E  DETAILS ON TRAINING AND COMPUTE

**Model Architecture:** We report the key comparisons with existing methods on two of the commonly considered model architectures in the literature of adversarial robustness Pang et al. (2021); Zhang et al. (2019); Rice et al. (2020); Croce et al. (2021) - ResNet-18 He et al. (2016) and WideResNet-34-10 Zagoruyko & Komodakis (2016). Although most existing methods for self-supervised adversarial training report results only on ResNet-18 Zhang et al. (2022); Fan et al. (2021), we additionally consider the WideResNet-34-10 architecture to demonstrate the scalability of the proposed approach to larger model architectures. We perform the ablation experiments on the CIFAR-100 dataset with WideResNet-34-10 architecture, which is a very challenging setting in self-supervised adversarial training, to be able to better distinguish between different variations adopted during training.

Table 9: **Total number of Forward (FP) or Backward (BP) Propagations during training** of the proposed approach when compared to prior works. Distillation based approaches - ProFeAT and DeACL require significantly lesser compute when compared to prior methods, and are only more expensive than supervised adversarial training.

| | # epochs | # attack steps | # FP or BP for AT | # FP or BP for auxiliary model | Total # FP or BP |
|---|---|---|---|---|---|
| Supervised (TRADES) | 110 | 10 | 1210 | 0 | 1210 |
| AP-DPE | 450 | 10 | 4950 | 0 | 4950 |
| RoCL | 1000 | 5 | 6000 | 0 | 6000 |
| ACL | 1000 | 5 | 12000 | 0 | 12000 |
| AdvCL | 1000 | 5 | 12000 | 1000 | 13000 |
| DynACL | 1000 | 5 | 12000 | 0 | 12000 |
| DynACL++ | 1025 | 5 | 12300 | 0 | 12300 |
| DeACL | 100 | 5 | 700 | 1000 | 1700 |
| ProFeAT (Ours) | 100 | 5 | 800 | 1000 | 1800 |

Table 10: **Floating Point Operations per Second (GFLOPS) and latency per epoch during training** of the proposed approach ProFeAT when compared to the baseline DeACL for ResNet-18 and WideResNet-34-10 models. The computational overhead during training is marginal with the addition of the projection layer, and reduces further for larger capacity models.

| | ResNet-18 | | | WideResNet-34-10 | | |
|---|---|---|---|---|---|---|
| | GFLOPS | Time/ epoch | #params (M) | GFLOPS | Time/ epoch | # params (M) |
| DeACL | 671827 | 51s | 11.27 | 6339652 | 4m 50s | 46.28 |
| ProFeAT (Ours) | 672200 | 51s | 11.76 | 6340197 | 4m 50s | 46.86 |
| % increase | 0.056 | 0.00 | 4.35 | 0.009 | 0.00 | 1.25 |

**Training Details:** The self-supervised training of the teacher model is performed for 1000 epochs with the SimCLR algorithm (Chen et al., 2020b) similar to prior work (Zhang et al., 2022). We utilize the solo-learn repository [1] for this purpose. For the SimCLR SSL training, we tune and use a learning rate of 1.5 with SGD optimizer, a cosine schedule with warmup, weight decay of $1e-5$ and train the backbone for 1000 epochs with other hyperparameters kept as default as in the repository.

The self-supervised adversarial training of the feature extractor using the proposed approach is performed for 100 epochs using SGD optimizer with a weight decay of $3e-4$, cosine learning rate with 10 epochs of warm-up, and a maximum learning rate of 0.5. We fix the value of $\beta$, the robustness-accuracy trade-off parameter (Ref: Eq.1 and 2 in the main paper) to 8 in all our experiments, unless specified otherwise.

**Details on Linear Probing:** To evaluate the performance of the learned representations, we perform standard linear probing by freezing the adversarially pretrained backbone as discussed in Section-2 of the main paper. We use a class-balanced validation split consisting of 1000 images from the train set and perform early-stopping during training based on the performance on the validation set. The training is performed for 25 epochs with a step learning rate schedule where the maximum learning rate is decayed by a factor of 10 at epoch 15 and 20. The learning rate is chosen amongst the following settings - $\{0.1, 0.05, 0.1, 0.5, 1, 5\}$ with SGD optimizer, and the weight decay is fixed to $2e-4$. The same evaluation protocol is used for the best baseline - DeACL Zhang et al. (2022) as well as the proposed approach, for both in-domain and transfer learning settings.

**Compute:** The following Nvidia GPUs have been used for performing the experiments reported in this work - V100, A100, and A6000. Each of the experiments are run either on a single GPU, or across 2 GPUs based on the complexity of the run and GPU availability. For 100 epochs of single-precision (FP32) training with a batch size of 256, the proposed approach takes ~8 hours and ~16GB of GPU memory on a single A100 GPU for WideResNet-34-10 model on CIFAR-100.

## F  COMPUTATIONAL COMPLEXITY

In terms of compute, both the proposed method ProFeAT and DeACL Zhang et al. (2022) lower the overall computational cost when compared to prior approaches. This is because self-supervised

---

[1]https://github.com/vturrisi/solo-learn

Table 11: **Diverse representations of Standard Trained (ST) and Adversarially Trained (AT) models (CIFAR-100, WRN-34-10):** ST models achieve 0% robust accuracy even with adversarial training of the linear layer, and AT models lose their robustness with standard full-finetuning. SA: Standard Accuracy, RA-G: Robust accuracy against GAMA, RA-PGD20: Robust Accuracy against PGD-20 attack

| Training/ LP Method | SA | RA-PGD20 | RA-G | Training/ LP Method | SA | RA-PGD20 | RA-G |
|---|---|---|---|---|---|---|---|
| Standard trained model | 80.86 | 0.00 | 0.00 | TRADES AT model | 60.22 | 28.67 | 26.36 |
| + Adversarial Linear Probing | 80.10 | 0.00 | 0.00 | + Standard Full Finetuning | 76.11 | 0.37 | 0.11 |

Table 12: **Failure of AD5 (Table-7) defense loss for SSL-AT training of WRN-34-10 model on CIFAR-100:** Using clean and adversarial loss exclusively at projector and feature space respectively, results in an unstable optimization problem, giving either a non-robust model or collapsed representations at the end of training. As shown below, a lower value of $\beta$ (the robustness-accuracy trade-off parameter) results in a non-robust model, while higher $\beta$ results in the learning of collapsed representations.

| $\beta$ | Standard Accuracy (SA) | Robust Accuracy against GAMA (RA-G) |
|---|---|---|
| 1 | **67.34** | 0.46 |
| 5 | 51.99 | 0.71 |
| 10 | 31.34 | 7.81 |
| 50 | 11.59 | 2.55 |
| 100 | 8.23 | **2.61** |

training in general requires larger number of epochs (1000) to converge Chen et al. (2020b); He et al. (2020) when compared to supervised learning (<100). Prior approaches like RoCL Kim et al. (2020), ACL Jiang et al. (2020) and AdvCL Fan et al. (2021) combine the contrastive training objective of SSL approaches and the adversarial training objective. Thus, these methods require larger number of training epochs (1000) for the adversarial training task, which is already computationally expensive due to the requirement of generating multi-step attacks during training. ProFeAT and DeACL use a SSL teacher for training and thus, the adversarial training is more similar to supervised training, requiring only 100 epochs. In Table-9, we present the approximate number of forward and backward propagations for each algorithm, considering both pretraining of the auxiliary network used and the training of the main network. It can be noted that the distillation based approaches - ProFeAT and DeACL require significantly lesser compute when compared to prior methods, and are only more expensive than supervised adversarial training. In Table-10, we present the FLOPS required during training for the proposed approach and DeACL. One can observe that there is a negligible increase in FLOPS compared to the baseline approach.

# G  ADDITIONAL RESULTS

## G.1  EFFECT OF EACH COMPONENT OF THE PROPOSED APPROACH

We show the impact of each component of the proposed approach in Table-13. Below are the observations based on the results:

- *Projector*: A key component of the proposed method is the introduction of the projector. We observe significant gains in clean accuracy (4.76%) by introducing the frozen pretrained projector along with defense losses at the feature and projector (E1 vs. E2). The importance of the projector is also evident by the fact that removing the projector from the proposed defense results in a large drop (5.7%) in clean accuracy (E9 vs. E5). In combination with other components in the table as well, the projector improves the performance significantly (E4 vs. E6, E3 vs. E7, E5 vs. E9). We observe an improvement of 9.18% in clean accuracy when the projector is introduced in the presence of the proposed augmentation strategy (E3 vs. E7). This is significantly higher than the gains obtained by introducing the projector over the baseline, which is 4.76% (E1 vs. E2).

- *Augmentations*: The proposed augmentation strategy is seen to improve performance, specifically robust accuracy, across all combinations. Introducing the proposed augmen-

Table 13: **Ablations on ProFeAT (CIFAR-100, WRN-34-10):** Performance (%) by enabling different components of the proposed approach. A tick mark in the Projector column means that a frozen pretrained projector is used for the teacher and student, with the defense loss being enforced at the feature and projector as shown in Eq.3. E1 represents the baseline or DeACL defense, and E9 represents the proposed defense or ProFeAT. E8*:Defense loss applied only at the projector. SA: Standard Accuracy, RA-PGD20: Robust Accuracy against PGD-20 attack, RA-G: Robust Accuracy against GAMA

| E# | Projector | Augmentations | Attack loss | SA | RA-PGD20 | RA-G |
|---|---|---|---|---|---|---|
| E1 | | | | 52.90 | 32.75 | 24.66 |
| E2 | ✓ | | | 57.66 | 31.14 | 25.04 |
| E3 | | ✓ | | 52.83 | 35.00 | 27.13 |
| E4 | | | ✓ | 51.80 | 31.37 | 24.77 |
| E5 | | ✓ | ✓ | 55.35 | 35.89 | **27.86** |
| E6 | ✓ | | ✓ | 56.57 | 30.54 | 25.29 |
| E7 | ✓ | ✓ | | **62.01** | 31.62 | 26.89 |
| E8 | ✓* | ✓ | ✓ | 59.65 | 33.03 | 26.90 |
| E9 | ✓ | ✓ | ✓ | 61.05 | 31.99 | 27.41 |

Table 14: **Ablations on Projector Configuration and Architecture (CIFAR-100, WRN-34-10):** Performance (%) obtained by varying the projector configuration (config.) and architecture (arch.). A non-linear projector effectively reduces the gap in clean accuracy between the teacher and student. A bottleneck architecture for the projector is worse than other variants. SA: Standard Accuracy, RA-PGD20: Robust Accuracy against PGD-20 attack, RA-G: Robust Accuracy against GAMA

| # | Projector config. | Projector arch. | Teacher SA | Student SA | Drop in SA | % Drop in SA | RA-PGD20 | RA-G |
|---|---|---|---|---|---|---|---|---|
| APA1 | No projector | - | 70.85 | 55.35 | 15.50 | 21.88 | 35.89 | **27.86** |
| APA2 | Linear layer | 640-256 | 68.08 | 53.35 | 14.73 | 21.64 | 35.57 | 27.47 |
| **Ours** | 2 Layer MLP | 640-640-256 | 70.85 | 61.05 | 9.80 | 13.83 | 31.99 | 27.41 |
| APA3 | 3 Layer MLP | 640-640-640-256 | 70.71 | 60.37 | 10.34 | 14.62 | 31.44 | 27.37 |
| APA4 | 2 Layer MLP | 640-640-640 | 69.88 | 61.24 | 8.64 | 12.36 | 31.88 | 27.36 |
| APA5 | 2 Layer MLP | 640-2048-640 | **70.96** | **61.76** | 9.20 | 12.97 | 29.53 | 26.66 |
| APA6 | 2 Layer MLP | 640-256-640 | 69.37 | 57.87 | 11.50 | 16.58 | 34.53 | 27.56 |

tation strategy in the base DeACL defense improves the robust accuracy by 2.47% (E1 vs. E3). The importance of the proposed augmentation strategy is also evident by the fact that in the absence of the same, there is a 4.48% drop in clean accuracy and 2.12% drop in robust accuracy (E9 vs. E6). Further, when combined with other components as well, the proposed augmentation strategy shows good improvements (E4 vs. E5, E2 vs. E7, E6 vs. E9).

- *Attack loss*: The proposed attack objective is designed to be consistent with the proposed defense strategy, where the goal is to enforce smoothness at the student in the feature space and similarity with the teacher in the projector space. As shown in Table-17, the proposed method is not very sensitive to different choices of attack losses. The impact of the attack loss in feature space can be seen in combination with the proposed augmentations, where we observe an improvement of 2.5% in clean accuracy alongside a marginal improvement in robust accuracy (E3 vs. E5). However, in presence of projector, the attack results in only marginal robustness gains, possibly because the clean accuracy is already high (E9 vs. E7).

- *Defense loss*: We do not introduce a separate column for defense loss as it is applicable only in the presence of the projector. We show the impact of the proposed defense losses in the last two rows (E8 vs. E9). The proposed defense loss improves the clean accuracy by 1.4% and robust accuracy marginally.

## G.2 ARCHITECTURE OF THE PROJECTION HEAD

In the proposed approach, we use the following 2-layer MLP projection head for both self-supervised pretraining of the teacher and adversarial training of the student:

- ResNet-18: 512-512-256
- WideResNet-18: 640-640-256

Table 15: **Ablations on the number of training epochs for the teacher SSL model (CIFAR-100, WRN-34-10):** Performance (%) obtained by varying the number of epochs for which the standard self-supervised teacher model is pretrained. Improvements in accuracy of the teacher result in corresponding gains in both standard and robust accuracy of the student. SA: Standard Accuracy, RA-PGD20: Robust Accuracy against PGD-20 attack, RA-G: Robust Accuracy against GAMA

| # epochs of PT | Teacher SA | Student SA | Drop in SA | % Drop in SA | RA-PGD20 | RA-G |
|---|---|---|---|---|---|---|
| 100 | 55.73 | 49.37 | 6.36 | 11.41 | 25.23 | 20.86 |
| 200 | 65.43 | 56.16 | 9.27 | 14.17 | 28.67 | 24.15 |
| 500 | 69.27 | 59.62 | 9.65 | 13.93 | 31.46 | 26.75 |
| 1000 | **70.85** | **61.05** | 9.80 | 13.83 | 31.99 | **27.41** |

In Table-14, we present results using different configurations and architectures of the projector. Firstly, the use of a linear projector (APA2) is similar to the case where projector is not used for student training (APA1), with $\sim 21\%$ drop in clean accuracy of the student with respect to the teacher. This improves to $12 - 17\%$ when a non-linear projector is introduced (APA3-APA6 and Ours). The use of a 2-layer MLP (Ours) is marginally better than the use of a 3-layer MLP (APA3) in terms of clean accuracy of the student. The accuracy of the student is stable across different architectures of the projector (Ours, APA4, APA5). However, the use of a bottleneck architecture (APA6) results in a higher drop in clean accuracy of the student.

### G.3 ACCURACY OF THE SELF-SUPERVISED TEACHER MODEL

The self-supervised teacher model is obtained using 1000 epochs of SimCLR (Chen et al., 2020b) training in all our experiments. We now study the impact of training the teacher model for lesser number of epochs. As shown in Table-15, as the number of teacher training epochs reduces, there is a drop in the accuracy of the teacher, resulting in a corresponding drop in the clean and robust accuracy of the student model. Thus, the performance of the teacher is crucial for training a better student model.

### G.4 SELF-SUPERVISED TRAINING ALGORITHM OF THE TEACHER

In the proposed approach, the teacher is trained using the popular self-supervised training algorithm SimCLR (Chen et al., 2020b), similar to prior works (Zhang et al., 2022). In this section, we study the impact of using different algorithms for the self-supervised training of the teacher and present results in Table-16. In order to ensure consistency across different SSL methods, we use a *random trainable* projector (2-layer MLP with both hidden and output dimensions of 640) for training the student and do not employ any projection head for the pretrained frozen teacher. While the default teacher trained using SimCLR was finetuned across hyperparameters, we utilize the default hyperparameters from the solo-learn[2] Github repository for this table, and thus present SimCLR also without tuning for a fair comparison. For uniformity, we report all results with $\beta = 8$ (the robustness-accuracy trade-off parameter). From Table-16, we note that in most cases, the clean accuracy of the student increases as the accuracy of the teacher improves, while the robust accuracy does not change much. We note that this table merely shows that the proposed approach can be effectively integrated with several base self-supervised learning algorithms for the teacher model. However, it does not present a fair comparison across different SSL pretraining algorithms, since the ranking on the final performance of the student would change if the pretraining SSL algorithms were used with appropriate hyperparameter tuning.

### G.5 ATTACK LOSS

For performing adversarial training using the proposed approach, attacks are generated by minimizing a combination of cosine similarity based losses as shown in Eq.4 of the main paper. This includes an unsupervised loss at the feature representations of the student and another loss between the representations of the teacher and student at the projector. As shown in Table-17, we obtain a

---

[2]https://github.com/vturrisi/solo-learn

Table 16: **Ablations on the algorithm used for training the self-supervised teacher model (CIFAR-100, WRN-34-10):** Performance (%) of the proposed approach by varying the pretraining algorithm of the teacher model. A random trainable projector is used for training the student model, to maintain uniformity in projector architecture across all methods. SA: Standard Accuracy, RA-PGD20: Robust Accuracy against PGD-20 attack, RA-G: Robust Accuracy against GAMA

| Teacher Training | Teacher SA | Student SA | RA-PGD20 | RA-G |
|---|---|---|---|---|
| SimCLR | 67.98 | 62.20 | 31.31 | 26.13 |
| SimCLR (tuned) | 70.85 | 63.07 | 32.05 | 26.57 |
| BYOL | **72.97** | 63.19 | 31.63 | **26.82** |
| Barlow Twins | 67.74 | 60.69 | 29.46 | 24.48 |
| SimSiam | 68.60 | 63.46 | 32.06 | 26.69 |
| MoCoV3 | 72.48 | **65.57** | 32.22 | 26.65 |
| DINO | 68.75 | 60.61 | 30.16 | 24.80 |

Table 17: **Ablations on Attack Loss (CIFAR-100, WRN-34-10):** Performance (%) with variations in attack loss at feature (feat.) and projector (proj.). While the proposed defense is stable across several variations in the attack loss, minimizing a combination of both losses $\cos(\mathcal{T}, \mathcal{S})$ and $\cos(\mathcal{S}, \mathcal{S})$ gives the best robustness-accuracy trade-off. SA: Standard Accuracy, RA-PGD20: Robust Accuracy against PGD-20 attack, RA-G: Robust Accuracy against GAMA

| # | Attack @ feat. | Attack @ proj. | SA | RA-PGD20 | RA-G |
|---|---|---|---|---|---|
| AT1 | $\cos(\mathcal{T}, \mathcal{S})$ | $\cos(\mathcal{T}, \mathcal{S})$ | 60.84 | 31.41 | 26.78 |
| AT2 | $\cos(\mathcal{S}, \mathcal{S})$ | $\cos(\mathcal{S}, \mathcal{S})$ | 61.30 | 31.86 | 26.75 |
| **Ours** | $\cos(\mathcal{S}, \mathcal{S})$ | $\cos(\mathcal{T}, \mathcal{S})$ | 61.05 | 31.99 | 27.41 |
| AT3 | $\cos(\mathcal{T}, \mathcal{S})$ | $\cos(\mathcal{S}, \mathcal{S})$ | 60.69 | 32.34 | **27.44** |
| AT4 | $\cos(\mathcal{S}, \mathcal{S})$ | - | 61.62 | 31.69 | 26.62 |
| AT5 | - | $\cos(\mathcal{S}, \mathcal{S})$ | 61.09 | 31.78 | 27.00 |
| AT6 | $\cos(\mathcal{T}, \mathcal{S})$ | - | **62.01** | 31.62 | 26.89 |
| AT7 | - | $\cos(\mathcal{T}, \mathcal{S})$ | 61.18 | 31.43 | 27.24 |

better robustness-accuracy trade-off by using a combination of both losses rather than by using only one of the two losses, due to better diversity and strength of attack. These results also demonstrate that the proposed method is not very sensitive to different choices of attack losses.

### G.6 WEIGHTING OF DEFENSE LOSSES AT THE FEATURE AND PROJECTOR

In the proposed approach, the defense losses are equally weighted between the feature and projector layers as shown in Eq.3 of the main paper. In Fig.4, we present results by varying the weighting $\lambda$ between the defense losses at the feature ($\mathcal{L}_f$) and projector ($\mathcal{L}_{fp}$) layers as shown below:

$$\mathcal{L}_{\text{ProFeAT}} = \lambda \cdot \mathcal{L}_f + (1 - \lambda) \cdot \mathcal{L}_{fp} \tag{5}$$

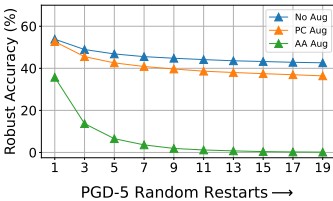

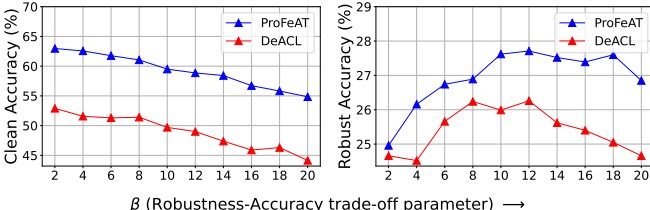

Figure 2: Robust accuracy of a supervised TRADES model across random restarts of PGD 5-step attack (CIFAR-100, WRN-34-10).

Figure 3: Performance of the proposed method ProFeAT when compared to DeACL (Zhang et al., 2022) across variation in the robustness-accuracy trade-off parameter $\beta$ on CIFAR-100 dataset with WRN-34-10 architecture.

Table 18: **Ablations on number of attack steps used for adversarial training (CIFAR-100, WRN-34-10):** Performance (%) using lesser number of attack steps (2 steps) when compared to the standard case (5 steps) during adversarial training. Clean/ Standard Accuracy (SA) and robust accuracy against GAMA (RA-G) and AutoAttack(RA-AA) are reported. The proposed approach is stable at lower attack steps as well, while being better than both TRADES (Zhang et al., 2019) and DeACL (Zhang et al., 2022).

| # attack steps | Supervised (TRADES) | | | DeACL | | | ProFeAT (Ours) | | |
|---|---|---|---|---|---|---|---|---|---|
| | SA | RA-G | RA-AA | SA | RA-G | RA-AA | SA | RA-G | RA-AA |
| 2 | **60.80** | 24.49 | 23.99 | 51.00 | 24.89 | 23.45 | 60.43 | 26.90 | **26.23** |
| 5 | **61.05** | 25.87 | 25.77 | 52.90 | 24.66 | 23.92 | **61.05** | 27.41 | **26.89** |

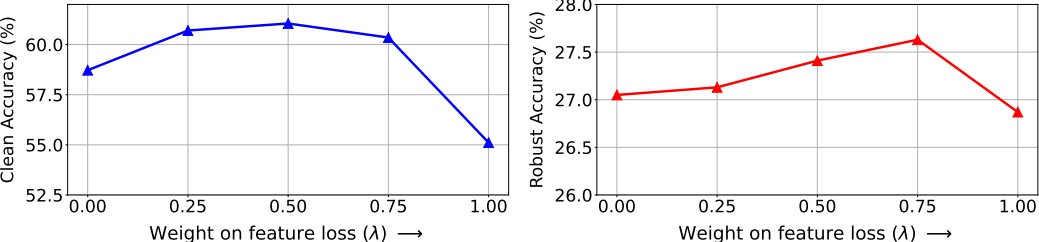

Figure 4: Performance (%) of the proposed approach ProFeAT by varying the weight between the defense losses at the feature and projector. A convex combination weighted by a factor $\lambda$ is used: $\mathcal{L}_{\mathrm{ProFeAT}} = \lambda \cdot \mathcal{L}_f + (1 - \lambda) \cdot \mathcal{L}_{fp}$, where $\mathcal{L}_{fp}$ and $\mathcal{L}_f$ represent the overall defense losses at the projector and feature respectively (Eq.1 and 2 of the main paper). The performance is stable across the range $\lambda \in [0.25, 0.75]$. We thus fix the value of $\lambda$ to $0.5$ in the proposed appraoch.

It can be noted that the two extreme cases of $\lambda = 0$ and $\lambda = 1$ result in a drop in clean accuracy, with a larger drop in the case where the loss is enforced only at the feature layer. The robust accuracy shows lesser variation across different values of $\lambda$. Thus, the overall performance is stable over the range $\lambda \in [0.25, 0.75]$, making the default setting of $\lambda = 0.5$ a suitable option.

### G.7    IMPROVING THE EFFICIENCY OF SELF-SUPERVISED ADVERSARIAL TRAINING

Similar to prior works (Zhang et al., 2022), the proposed approach uses 5 step PGD based optimization for attack generation during adversarial training. In Table-18, we present results with lesser optimization steps (2 steps). The proposed approach is stable and obtains similar results even by using 2-step attack. Even in this case, the clean and robust accuracy of the proposed approach is significantly better than the baseline approach DeACL (Zhang et al., 2022), and also outperforms the supervised TRADES model (Zhang et al., 2019).

### G.8    ADDITION EVALUATION ON SELF-SUPERVISED TRAINED MODELS

We compare the performance of DeACL (best baseline) and ProFeAT (Ours) on CIFAR-10 and CIFAR-100 datasets with WideResNet-34-10 architecture in Table-19. The model is first pretrained using the respective self-supervised adversarial training algorithm, and further we compute the standard accuracy (SA) and robust accuracy against GAMA (RA-G) using several methods such as standard linear probing (LP), training a 2 layer MLP head (MLP), and performing KNN in the feature space (k=10). We note that the proposed method achieves improvements over the baseline across all evaluation methods. Since the training of classifier head in LP and MLP is done using standard training and not adversarial training, the robust accuracy reduces as the number of layers increases (from linear to 2-layers), and the standard accuracy improves. The standard accuracy of KNN is better than the standard accuracy of LP for the baseline, indicating that the representations are not linearly separable. Whereas, as is standard, for the proposed approach, LP standard accuracy is higher than that obtained using KNN. The adversarial attack used for evaluating the robust accuracy using KNN is generated using GAMA attack on a linear classifier. The attack is suboptimal since it is not generated by using the evaluation process (KNN), and thus the robust accuracy against such an attack is higher.

Table 19: **Additional evaluation on pretrained models:** Performance (%) of DeACL (best baseline) and ProFeAT (Ours) on CIFAR-10 and CIFAR-100 datasets with WideResNet-34-10 architecture. The model is first pretrained using the respective self-supervised adversarial training algorithm, and further we compute the standard accuracy (SA) and robust accuracy against GAMA (RA-G) using several methods such as standard linear probing (LP), training a 2 layer MLP head (MLP), and performing KNN in the feature space (k=10). The proposed method achieves improvements over the baseline across all evaluation methods

|  | SA (LP) | RA-G (LP) | SA (MLP) | RA-G (MLP) | SA (KNN) | RA-G (KNN) |
|---|---|---|---|---|---|---|
| | | | **CIFAR-10** | | | |
| DeACL | 83.60 | 49.62 | 85.66 | 48.74 | 87.00 | 54.58 |
| ProFeAT | **87.44** | **52.24** | **89.37** | **50.00** | **87.38** | **55.77** |
| | | | **CIFAR-100** | | | |
| DeACL | 52.90 | 24.66 | 55.05 | 22.04 | 56.82 | 31.26 |
| ProFeAT | **61.05** | **27.41** | **63.81** | **26.10** | **58.09** | **32.26** |

Table 20: **Transfer Learning to STL-10 and Caltech-101:** Transfer learning performance (%) with adversarial full finetuning (AFF) using TRADES Zhang et al. (2019) algorithm for 25 epochs, from CIFAR-10 and CIFAR-100 to STL-10 and Caltech-101 datasets on WideResNet-34-10 architecture. The proposed method outperforms both DeACL and the supervised trained model. Standard Accuracy (SA), robustness against PGD-20 (RA-PGD20) and GAMA (RA-G) are reported.

|  | CIFAR-10 ->STL10 | | | CIFAR-100 ->STL10 | | |
|---|---|---|---|---|---|---|
|  | SA | RA-PGD20 | RA-G | SA | RA-PGD20 | RA-G |
| Supervised (TRADES) | 64.58 | 39.83 | 32.78 | 64.22 | 34.20 | 31.01 |
| DeACL | 61.65 | 31.88 | 28.34 | 60.89 | 33.06 | 30.00 |
| ProFeAT | **74.12** | 40.15 | **36.04** | **68.77** | 35.35 | **31.23** |
|  | CIFAR-10 ->Caltech-101 | | | CIFAR-100 ->Caltech-101 | | |
| Supervised (TRADES) | 62.46 | 40.77 | 39.40 | **64.97** | 42.95 | 41.02 |
| DeACL | 62.65 | 41.39 | 39.18 | 61.01 | 40.56 | 39.09 |
| ProFeAT | **66.11** | 45.29 | **42.12** | 64.16 | 42.95 | **41.25** |

## G.9 Transfer Learning using Adversarial Full Finetuning (AFF)

We present transfer learning results to STL-10 and Caltech-101 using lightweight adversarial full finetuning (AFF) in Table-20. Caltech-101 contains 101 object classes and 1 background class, with 2416 samples in the train set and 6728 samples in the test set. The number of samples per class range from 17 to 30, and thus this is a suitable dataset to highlight the practical importance of adversarial self-supervised pretrained representations for low-data regime. Towards this, a base robustly pretrained model is finetuned using the TRADES adversarial training for 25 epochs. We present results on WideResNet-34-10 models that are pretrained on CIFAR-10 and CIFAR-100 respectively. We note that the proposed method outperforms DeACL by a large margin. Further, we note that by using merely 25 epochs of adversarial finetuning using TRADES, the proposed method achieves improvements of around 4% on CIFAR-10 and 11% on CIFAR-100 when compared to the linear probing accuracy presented in Table-5, highlighting the practical utility of the proposed method. The AFF performance of the proposed approach is better than that of a supervised TRADES pretrained model as well.

