# OpenReview forum: "ProFeAT: Projected Feature Adversarial Training for Self-Supervised Learning of Robust Representations"
_ICLR.cc/2024/Conference — Submitted to ICLR 2024_

### Official Review · Reviewer_eRyw · 2023-11-01

**Soundness:** 3 good
**Presentation:** 4 excellent
**Contribution:** 3 good
**Rating:** 6
**Confidence:** 5

**Summary:**

This paper proposed to bridge the gap between the self-supervised and supervised adversarial training methods, with good scalability for larger models.

This paper is well-written and easy to follow.

**Strengths:**

The topic of the paper, bridging the gap between supervised and self-supervised adversarial training, is interesting.

The result is convincing.

This paper applied the proposed method to different DNN models, including popular VIT.

Extensive ablation studies are provided, which shows the insight of the proposed method.

Extensive results are shown in the appendices, which are helpful for the readers to understand the whole story.

**Weaknesses:**

First, the author should give some basic explanation for the results, which will be appreciated. For example:
1. In Table 3, the author should at least explain what is "SA" (standard accuracy?).
2. In Table 4, the author should at least highlight which method has better results for each network structure (each row).
3. what is the adversarial perturbation upper bound for testing in the table 3 and 4?

In Table 3, for DeACL (reproduced), the author modified the original teacher model. In this case, the word "reproduced" is misleading. In fact. it is not "reproduced", but modified.

"DynACL is run for 500 epochs rather than....":
In this case, the reported result of DynACL should not be used for comparison. Because the result is not from the proposed settings, the result is not convincing. The explanation in Appendix D cannot overcome this problem.

**Questions:**

1. In Table 3, why not highlight the best result in column PGD-20?

2. According to the results, the proposed method gives a better improvement on the larger model (WRN-34-10) than the smaller one (RES-18). This is very interesting. Can the author give some explanation and insight about this?

---

> ### Author Response · Authors · 2023-11-15
> **Response to reviewer eRyw**
>
> We thank the reviewer for their encouraging comments and valuable feedback. We clarify their concerns below and upload a new version of the paper incorporating their feedback.
>
> - Clarifications on tables:
>     - Yes, SA stands for standard accuracy in the tables. We mention this in the captions of all tables in the updated draft for better clarity.
>     - In the updated draft, we bold the best results in each row of Table-4 as suggested.
>     - Our training and evaluation uses an $\ell_\infty$ perturbation bound of 8/255 as mentioned in Section-2.
>     - We update the terminology used for DeACL results with our teacher model from "Reproduced" to "Our Teacher" for better clarity.
> - DynACL: While initially we presented only 500 epoch result on the WideResNet-34-10 architecture due to limited compute, we have later verified that our method outperforms even the 1000 epoch run. We present results of a 1000 epoch run for DynACL++ on the WideResNet-34-10 architecture below:
>
> |                        |  |   **CIFAR-10**           |                        | |     **CIFAR-100**          |                         |
> |------------------------|:------------:|:------------:|:----------------------:|:-------------:|:------------:|:----------------------:|
> |                        |    **SA**    | **RA-PGD20** | **RA-AA (AutoAttack)** |     **SA**    | **RA-PGD20** | **RA-AA (AutoAttack)** |
> |  DynACL++ (500 epochs) |     82.27    |     49.60    |          47.12         |     52.59     |     24.22    |          21.27         |
> | DynACL++ (1000 epochs) |     80.97    |     48.28    |          45.50         |     52.60     |     23.42    |          20.58         |
> |          DeACL         |  83.83  |  57.09  |       48.85       |   52.92 |  32.66  |       23.82       |
> |     ProFeAT (Ours)     |  **87.62**  |  54.50 |       **51.95**       |   **61.08**  |  31.96  |       **26.81**       |
>
>
> We note that the 500 epoch DynACL++ run gives better results, possibly due to robust overfitting in a 1000 epoch run. We added these results in Table-3 of the updated draft.
>
>
>   - Highlighting the best result in column PGD-20:
>      -  The gap between the true robustness of a model (which can be approximated by accuracy against AutoAttack or GAMA) and the robustness against an attack such as PGD-20 (with a small number of optimization steps) depends on the local linearity of the loss surface. For a perfectly linear loss surface, a single-step attack such as FGSM can also find the adversary effectively. However, for more convoluted loss surfaces, multiple attack steps with adaptively reducing step size and better loss functions are required to find the true loss maxima. Thus, a lower gap between PGD-20 and AutoAttack implies better smoothness of the loss surface.
>      - Let us consider a case where the proposed method has better robust accuracy against AutoAttack and worse accuracy against PGD-20 (such as DeACL vs. Ours in Table-3). This happens because the proposed method has a smoother loss surface (due to better robustness) and thus PGD 20 step attack is able to reliably find the adversarial attack that exists. Hence, for a given value of Robust Accuracy against AutoAttack, a lower value of PGD-20 is better than having a higher value. Thus, we believe it is misleading to highlight the best PGD-20 accuracy. We clarify this in Section-2 of the updated draft.
>
>   - Better results on larger models: For a sufficiently complex task, a scalable approach should result in better performance on larger models given enough data. Although the task complexity of adversarial self-supervised learning is high, the gains in prior approaches are marginal with an increase in model size, while the proposed method results in significantly improved performance on larger capacity models. We discuss the key factors that result in better scalability below:
>     - As discussed in Section-4.1, a mismatch between training objectives of the teacher and ideal goals of the student causes a drop in student performance. This primarily happens because of the overfitting to the teacher training task. As model size increases, the extent of overfitting increases. The use of a projection layer during distillation alleviates the impact of this overfitting and allows the student to retain more generic features that are useful for the downstream robust classification objective. Thus, a projection layer is more important for larger model capacities where the extent of overfitting is higher.
>     - Secondly, as the model size increases, there is a need for higher amount of training data for achieving better generalization. The proposed method has better data diversity as it enables the use of more complex data augmentations in adversarial training by leveraging supervision from weak augmentations at the teacher.
>
> We hope this clarifies the concerns of the reviewer. We will be happy to answer any further questions as well.

---

### Official Review · Reviewer_e5nc · 2023-11-01

**Soundness:** 2 fair
**Presentation:** 2 fair
**Contribution:** 2 fair
**Rating:** 5
**Confidence:** 4

**Summary:**

The paper introduces a novel self-supervised learning (SSL) method named ProFeAT to enhance the robustness of Deep Neural Networks (DNNs) against adversarial attacks. While supervised adversarial training has proven effective, it demands extensive labeled data, leading to high costs. Previous SSL attempts, including SimCLR and Decoupled Adversarial Contrastive Learning (DeACL), have shown limitations in performance and increased training complexity, particularly with larger models. ProFeAT addresses these issues by incorporating a projection head in the adversarial training step, defining specific attack and defense losses, and employing a mix of weak and strong augmentations for the teacher-student setting. This strategy aims to close the performance gap between self-supervised and supervised adversarial training, enhancing generalization without adding to the training complexity.

**Strengths:**

1. ProFeAT introduces a novel method to improve the robustness of DNNs through self-supervised learning, addressing the challenges of previous SSL adversarial training methods.
2. The paper provides extensive experimental results, demonstrating the effectiveness of ProFeAT across different datasets and model architectures.

**Weaknesses:**

1. The paper primarily relies on linear probing for evaluation, which is just one of several methods to assess the quality of a trained encoder. It is crucial to explore alternative evaluation techniques such as K-nearest neighbors (KNN) to validate the model's performance comprehensively. Additionally, the effectiveness of the pretraining method in downstream tasks with the finetuning method should be rigorously verified.

2. Section 4.1 lacks compelling evidence and in-depth analysis. The paper lacks a thorough explanation of **why** objective matching leads to improved performance, and there is insufficient exploration of **how** aligning the linear probing objective with pretraining aids in distillation. The correlation between high cosine similarity and low performance is demonstrated in both Table 1, and 2 but lacks meaningful context.

3. While the empirical exploration of simple/difficult augmentation combinations is commendable, the paper lacks a robust analysis or rationale behind why the proposed combinations are deemed the most effective.

4. Although the paper explores various approaches, the methodology lacks strong justification, making it challenging to establish the credibility of the proposed methods. A more thorough and convincing demonstration of these approaches is needed.

**Questions:**

In my view, this paper demonstrates that the use of the freeze teacher projector can provide a slightly more robust constraint, promoting alignment between the student's feature space and that of the teacher, thus facilitating stable student learning. Consequently, the results show only marginal improvements compared to DeACL for smaller models, while proving more effective in scenarios where distillation regularization becomes challenging, particularly as the model scales up in size. I wonder what the authors think about this interpretation.

---

> ### Author Response · Authors · 2023-11-17
> **Response to reviewer e5nc [1/3]**
>
> We thank the reviewer for their insightful comments and valuable feedback. We clarify their concerns below.
>
> 1. **Additional methods for evaluating the pretrained models**: As discussed in Section-2, we primarily rely on the standard linear probing evaluation, as freezing the backbone can give a more reliable estimate of the robustness of the representations learned when compared to a finetuning framework that can alter the representations significantly. However, we agree that other types of evaluations such as KNN and finetuning can be valuable in comprehensively comparing the nature of representations learned. Therefore, as suggested, we present below the performance of the proposed method when compared to the best baseline DeACL on evaluation metrics such as KNN and Adversarial Full Finetuning (AFF):
>  - We compare the performance of DeACL (best baseline) and ProFeAT (Ours) on CIFAR-10 and CIFAR-100 datasets with WideResNet-34-10 architecture. The model is first pretrained using the respective self-supervised adversarial training algorithm, and further we compute the standard accuracy (SA) and robust accuracy against GAMA (RA-G) using several methods such as standard linear probing (LP), training a 2 layer MLP head (MLP), and performing KNN in the feature space (k=10).
>
> |       CIFAR-10      | SA (LP) | RA-G (LP) | SA (MLP) | RA-G (MLP) | SA (KNN) | RA-G (KNN) |
> |:-------------------:|:-------:|:---------:|:--------:|:----------:|:--------:|:----------:|
> |        DeACL        |  83.60  |   49.62   |   85.66  |    48.74   |   87.00  |    54.58   |
> |       ProFeAT       |  87.44  |   52.24   |   89.37  |    50.00   |   87.38  |    55.77   |
>
> |      CIFAR-100      | SA (LP) | RA-G (LP) | SA (MLP) | RA-G (MLP) | SA (KNN) | RA-G (KNN) |
> |:-------------------:|:-------:|:---------:|:--------:|:----------:|:--------:|:----------:|
> |        DeACL        |  52.90  |   24.66   |   55.05  |    22.04   |   56.82  |    31.26   |
> |       ProFeAT       |  61.05  |   27.41   |   63.81  |    26.10   |   58.09  |    32.26   |
>
>   -  We note that the proposed method achieves improvements over the baseline across all evaluation methods.
>       - Since the training of classifier head in LP and MLP is done using standard training and not adversarial training, the robust accuracy reduces as the number of layers increases (from linear to 2-layers), and the standard accuracy improves.
>       - The standard accuracy of KNN is better than the standard accuracy of LP for the baseline, indicating that the representations are not linearly separable. Whereas, as is standard, for the proposed approach, LP standard accuracy is higher than that obtained using KNN.
>       - The adversarial attack used for evaluating the robust accuracy using KNN is generated using GAMA attack on a linear classifier. The attack is suboptimal since it is not generated by using the evaluation process (KNN), and thus the robust accuracy against such a weak attack is higher.
>
>   - We next present the Adversarial Full Finetuning (AFF) performance on the downstream task of transfer learning from CIFAR-10/ CIFAR-100 to STL-10 and Caltech-101 respectively on WideResNet-34-10 architecture. AFF is performed using TRADES algorithm for 25 epochs.
>
> |                     | | CIFAR-10 -> STL10          |       |  |  CIFAR-100 -> STL10        |       |
> |---------------------|:-----------------:|:--------:|:-----:|:------------------:|:--------:|:-----:|
> |          |         **SA**        | **RA-PGD20** |  **RA-G** |         **SA**         | **RA-PGD20** |  **RA-G** |
> | Supervised (TRADES) |       64.58       |   39.83  | 32.78 |        64.22       |   34.20  | 31.01 |
> |        DeACL        |      61.65	|31.88	|28.34	|60.89	|33.06	|30.00|
> |       ProFeAT       |       74.12       |   40.15  | 36.04 |        68.77       |   35.35  | 31.23 |
>
> |                     |  | CIFAR-10 -> Caltech101         |       | |  CIFAR-100 -> Caltech101         |       |
> |---------------------|:----------------------:|:--------:|:-----:|:-----------------------:|:--------:|:-----:|
> |          |           **SA**           | **RA-PGD20** |  **RA-G** |            **SA**           | **RA-PGD20** |  **RA-G** |
> | Supervised (TRADES) |    62.46    |   40.77  | 39.40 |          64.97          |   42.95  | 41.02 |
> |        DeACL        |   62.65    |   41.39  | 39.18 |   61.01   |   40.56  | 39.09 |
> |       ProFeAT (Ours)      |   66.11 |   45.29  | 42.12 |    64.16      |   42.95  | 41.25 |
>
>    -  The proposed method outperforms DeACL by a large margin. Further, by using merely 25 epochs of adversarial finetuning, the proposed method achieves improvements of around 4% on CIFAR-10 and 11% on CIFAR-100 when compared to the linear probing accuracy presented in Table-5, highlighting the practical utility of the proposed method. The AFF performance of the proposed approach is better than that of a supervised TRADES pretrained model as well.
>
>                      (response continued in the next comment)

---

> > ### Author Response · Authors · 2023-11-17
> > **Response to reviewer e5nc [2/3]**
> >
> > 2. **Why and how in Section 4.1**: Based on the findings in prior work, we aim to clarify the intuition behind our hypotheses below:
> >   -  Why objective matching between teacher and student leads to improved performance: Let us consider task-A to be the teacher's training task, and task-B to be the student's downstream task or its ideal goal. The representations in deeper layers (last few layers) of the teacher are more tuned to its training objective (task-A), and the early layers contain a lot more information than what is needed for this task [1]. Thus, when the teacher is trained for a given task (task-A), features that are specific to that task are dominant in the final feature layer, and other features that may be relevant to task-B are sparse. As discussed in [2], this results in replication of features that are relevant to the task. When a similarity based distillation loss is enforced on such features, higher importance is given to matching the replicated features, and the sparse features which may be important for task-B are suppressed further in the student. On the other hand, when the student's task matches with the teacher's task (task-A = task-B), a similarity based distillation loss is very effective in transferring the necessary representations to the student, since they are predominant. Thus, matching the training objective of the teacher with the ideal goals of the student improves downstream performance.
> >    - How aligning the linear probing objective with pretraining aids in distillation: For a given network, aligning the pretraining task with downstream task results in better performance since the matching of tasks ensures that the required features are predominant, and they are easily used by an SVM classifier (or a linear classifier) trained over it (Claim 2.1 in [2]). In context of distillation, since the features of the student are trained by enforcing similarity based loss w.r.t. the teacher, we hypothesize that enforcing similarity w.r.t. the teacher is the best way to learn a student classifier as well. Let us consider task-A to be the teacher pretraining task, and task-B to be the downstream task or ideal goal of the student. As discussed above, the teacher's features are aligned to task-A and these are transferred effectively to the student. The features related to task-B are suppressed in the teacher and are further suppressed in the student. As the features specific to a given task become more sparse, it is harder for an SVM classifier (or a linear classifier) to rely on that feature, although it important for classification [2]. Thus, training a linear classifier for task-B is more effective on the teacher when compared to the student. It is to be noted that although task-B features exist in the student as well, the downstream classifier is unable to utilize it since they are sparse. The linear classifier of the teacher in effect *amplifies* the sparse features, allowing the student to learn them more effectively. So, learning a classifier on the teacher and distilling it to the student is better than training a classifier directly on the student.
> >
> >
> > [1] Florian Bordes, Randall Balestriero, Quentin Garrido, Adrien Bardes, and Pascal Vincent. Guillotine regularization: Improving deep networks generalization by removing their head. TMLR 2023.
> >
> > [2] Addepalli, Sravanti, Anshul Nasery, Venkatesh Babu Radhakrishnan, Praneeth Netrapalli, and Prateek Jain. "Feature Reconstruction From Outputs Can Mitigate Simplicity Bias in Neural Networks." ICLR 2023.
> >
> >        (response continued in the next comment)

---

> > > ### Author Response · Authors · 2023-11-17
> > > **Response to reviewer e5nc [3/3]**
> > >
> > > 3. & 4. **Overall justification of the proposed method/ Why the proposed combination of augmentations is most effective**: For a sufficiently complex task, a scalable approach should result in better performance on larger models given enough diverse data. Although the task complexity of adversarial self-supervised learning is high, the gains in prior approaches are marginal with an increase in model size (Table-3). Towards improving the scalability to larger models, we propose the following:
> > >  - *Projection layer along with appropriate attack/defense losses:* As discussed in Section-4.1, a mismatch between training objectives of the teacher and ideal goals of the student causes a drop in student performance. This primarily happens because of the overfitting to the teacher training task (more details in the response to Q-2 in the above comment). As model size increases, the extent of overfitting increases. The use of a projection layer during distillation alleviates the impact of this overfitting and allows the student to retain more generic features that are useful for the downstream robust classification objective. Thus, a projection layer is more important for larger model capacities where the extent of overfitting is higher.
> > >  - *Augmentation strategy:* Secondly, as the model size increases, there is a need for higher amount of training data for achieving better generalization. Towards this, we propose to leverage augmentations such as AutoAugment, which is known to improve generalization in a standard training regime. However, the effective use of augmentations in supervised and self-supervised adversarial training has been very challenging due to increased training complexity, and it is common to use the basic pad+crop augmentation alone for adversarial training [3-7]. Thus, although augmentations at the student model increase the data and attack diversity (as shown in Fig.2), they also increase the task complexity  and the domain shift between the train and test distribution, resulting in degradation in standard and robust accuracy. We aim to mitigate these concerns by introducing weak augmentations that are close to the test distribution during training. While the use of only strong augmentations expects the student to learn invariant representations by itself making the training task complex, the proposed method makes the task easier by mapping the representation of every strong augmentation to the representation of a weak augmentation that is close to the unaugmented image. Thus, strong augmentations at the student increase the data and attack diversity, while weak augmentations at the teacher provide better supervision and alleviate the task complexity.
> > >
> > > [3] Addepalli, Sravanti, Samyak Jain, and Venkatesh Babu Radhakrishnan. Efficient and effective augmentation strategy for adversarial training. NeurIPS 2022
> > >
> > > [4] L. Rice, E. Wong, and J. Z. Kolter. Overfitting in adversarially robust deep learning. ICML 2020
> > >
> > > [5] S. Gowal, C. Qin, J. Uesato, T. Mann, and P. Kohli. Uncovering the limits of adversarial training against norm-bounded adversarial examples. arXiv preprint arXiv:2010.03593, 2020.
> > >
> > > [6] H. Zhang, Y. Yu, J. Jiao, E. Xing, L. El Ghaoui, and M. I. Jordan. Theoretically principled trade-off between robustness and accuracy. ICML 2019
> > >
> > > [7] Chaoning Zhang, Kang Zhang, Chenshuang Zhang, Axi Niu, Jiu Feng, Chang D Yoo, and In So Kweon. Decoupled adversarial contrastive learning for self-supervised adversarial robustness. ECCV 2022
> > >
> > > 5. **The reviewer's hypothesis on why frozen projector head improves student performance**: It is indeed very compelling to believe that the reason for improved performance is the presence of a more robust constraint due to the frozen projector head, and a reduction in feature dimension, leading to higher improvements in larger models. However, we empirically find that these are not the main reasons for improved performance. We detail our findings below:
> > > - As shown in Table-6 of the main paper (AP2), even a random trainable projection head on the student improves the performance significantly, although the use of the frozen projection head from the teacher results in further improvements. Thus, the key insight is in using a projection head for the student model, rather than reusing the projection head from pretraining.
> > > - As shown in Table-13 (APA4), even by maintaining the same dimension of feature space before and after the projection head (640-640-640), we obtain similar results as the proposed approach (640-640-256). Thus the performance gains are not a result of the reduction in feature dimensions that can provide in a better constraint during training.
> > > - Our explanation of why the projector improves student performance is detailed in Section-4.1 and the responses above.
> > >
> > > We hope the rebuttal is able to clarify the reviewer's concerns, and we will be happy to answer any further questions as well.

---

### Official Review · Reviewer_s59e · 2023-11-06

**Soundness:** 3 good
**Presentation:** 3 good
**Contribution:** 3 good
**Rating:** 6
**Confidence:** 5

**Summary:**

The paper proposes Projected Feature Adversarial Training (ProFeAT) to address the gap between the target of self-supervised training and adversarial training in the teacher-student distillation setting. ProFeAT uses a frozen pretrained projection head from the teacher to isolate impact of distillation loss and prevent the overfitting of the student to the teacher training objective. The performance of ProFeAT is better than existing SSL adversarial training method especially on larger models.

**Strengths:**

The paper is clearly written.

The experimental result is solid, including models of different sizes and extensive ablation study.

The performance of ProFeAT is competitive on large-scale models.

**Weaknesses:**

1. There are multiple components in the proposed method, including an additional loss of projection layers, a new attack generation method and weak data augmentation. Although the ablation study includes variants of each component, it is still unknown the specific contribution of each component, or the combination of any two components. A mechanism for why the combination of the three components works is needed.

2. The experiment is done on CIFAR10 and CIFAR100, which contains sufficient data to train adversarially robust models even without distillation and pre-trained models [1]. It is more interesting to see the performance of ProFeAT on low-data tasks such as Caltech [2] as [3] shows that the benefit of pre-trained models mainly manefests in low-data tasks.

[1] Wu, Dongxian, Shu-Tao Xia, and Yisen Wang. "Adversarial weight perturbation helps robust generalization." Advances in Neural Information Processing Systems 33 (2020): 2958-2969.
[2] Griffin, Gregory, Alex Holub, and Pietro Perona. "Caltech-256 object category dataset." (2007).
[3] Liu, Ziquan, et al. "TWINS: A Fine-Tuning Framework for Improved Transferability of Adversarial Robustness and Generalization." Proceedings of the IEEE/CVF Conference on Computer Vision and Pattern Recognition. 2023.

**Questions:**

1. Could the authors show the effect of each component in the proposed method?

2. It is more convincing to show the effectiveness of the proposed method on small datasets such as Caltech. I believe the benefit of using self-supervised pre-training would be significant on such small datasets.

---

> ### Author Response · Authors · 2023-11-18
> **Response to reviewer s59e [1/2]**
>
> We thank the reviewer for their encouraging comments and valuable feedback. We present our responses to their questions below.
>
>
> 1. **Effect of each component in the proposed method:** While we present extensive ablation experiments for each component of the proposed method, we agree with the reviewer that one table that summarizes it all, and shows the impact of each component/ combination of any two components is a valuable addition to our work. We present a table that shows this below (SA:Standard Accuracy or Clean accuracy, RA-PGD20: Robust Accuracy against PGD-20, RA-G: Robust Accuracy against GAMA):
>
> |E#| Projector | Augmentations | Attack loss  |   SA  | RA-PGD20 |  RA-G |                Comments               |
> |:--:|:---------:|:---------------:|:------------:|:-----:|:--------:|:-----:|-------------------------------------|
> |E1|           |               |              | 52.90 |   32.75  | 24.66 | Baseline (DeACL)                      |
> |E2|     ✔     |               |              | 57.66 |   31.14  | 25.04 |                                       |
> |E3|           |       ✔       |              | 52.83 |   35.00  | 27.13 |                                       |
> |E4|           |               |       ✔      | 51.80 |   31.37  | 24.77 |                                       |
> |E5|           |       ✔       |       ✔      | 55.35 |   35.89  | **27.86** |                                       |
> |E6|     ✔     |               |       ✔      | 56.57 |   30.54  | 25.29 |                                       |
> |E7|     ✔     |       ✔       |              | **62.01** |   31.62  | 26.89 |                                       |
> |E8|      ✔    |       ✔       |       ✔      | 59.65 |   33.03  | 26.90| Defense loss only at the projector  |
> |E9|     ✔     |       ✔       |       ✔      | 61.05 |   31.99  | 27.41 | ProFeAT (Ours) |
>
>  - A tick mark in the *Projector* column imples that a frozen pretrained projector is used for the teacher and student, with the defense loss being enforced at the feature and projector as shown in Eq.3 of the main paper.
>
>  -  E1 represents the baseline or DeACL defense, and E9 represents the proposed defense or ProFeAT.
>
>  - *Projector*: A key component of the proposed method is the introduction of the projector. We observe significant gains in clean accuracy (4.76%) by introducing the frozen pretrained projector along with defense losses at the feature and projector (E1 vs. E2). The importance of the projector is also evident by the fact that removing the projector from the proposed defense results in a large drop (5.7%) in clean accuracy (E9 vs. E5). In combination with other components in the table as well, the projector improves the performance significantly (E4 vs. E6, E3 vs. E7, E5 vs. E9). We observe an improvement of 9.18% in clean accuracy when the projector is introduced in the presence of the proposed augmentation strategy (E3 vs. E7). This is significantly higher than the gains obtained by introducing the projector over the baseline, which is 4.76% (E1 vs. E2).
>
>  - *Augmentations*: The proposed augmentation strategy is seen to improve performance, specifically robust accuracy, across all combinations. Introducing the proposed augmentation strategy in the base DeACL defense improves the robust accuracy by 2.47% (E1 vs. E3). The importance of the proposed augmentation strategy is also evident by the fact that in the absence of the same, there is a 4.48% drop in clean accuracy and 2.12% drop in robust accuracy (E9 vs. E6). Further, when combined with other components as well, the proposed augmentation strategy shows good improvements (E4 vs. E5, E2 vs. E7, E6 vs. E9).
>
>  - *Attack loss*: The proposed attack objective is designed to be consistent with the proposed defense strategy, where the goal is to enforce smoothness at the student in the feature space and similarity with the teacher in the projector space. As shown in Table-16, the proposed method is not very sensitive to different choices of attack losses. The impact of the attack loss in feature space can be seen in combination with the proposed augmentations, where we observe an improvement of 2.5% in clean accuracy alongside a marginal improvement in robust accuracy (E3 vs. E5). However, in presence of projector, the attack results in only marginal robustness gains, possibly because the clean accuracy is already high (E9 vs. E7).
>
>  - *Defense loss*: We do not introduce a separate column for defense loss as it is applicable only in the presence of the projector. We show the impact of the proposed defense losses in the last two rows (E8 vs. E9). The proposed defense loss improves the clean accuracy by 1.4% and robust accuracy marginally.
>
>           (response continued in the next comment)

---

> ### Author Response · Authors · 2023-11-18
> **Response to reviewer s59e [2/2]**
>
> - *A mechanism for why the combination is needed*: For a sufficiently complex task, a scalable approach should result in better performance on larger models given enough data. Although the task complexity of adversarial self-supervised learning is high, the gains in prior approaches are marginal with an increase in model size, while the proposed method results in significantly improved performance on larger capacity models. We discuss the key factors that result in better scalability below:
>     - As discussed in Section-4.1, a mismatch between training objectives of the teacher and ideal goals of the student causes a drop in student performance. This primarily happens because of the overfitting to the teacher training task. As model size increases, the extent of overfitting increases. The use of a projection layer during distillation alleviates the impact of this overfitting and allows the student to retain more generic features that are useful for the downstream robust classification objective. Thus, a projection layer is more important for larger model capacities where the extent of overfitting is higher.
>     - Secondly, as the model size increases, there is a need for higher amount of training data for achieving better generalization. The proposed method has better data diversity as it enables the use of more complex data augmentations by leveraging supervision from weak augmentations at the teacher.
>
> 2. **Performance on small datasets**: We agree with the reviewer that it is important to show the transfer learning performance on small datasets. Towards this, we show the transfer learning performance from CIFAR-10/CIFAR-100 to STL-10 with linear probing in Table-5 of the main paper. However, in practice it is common to finetune the full network for better downstream performance. We thus present transfer learning results to STL-10 and Caltech-101 using lightweight adversarial full finetuning (AFF).
> Caltech-101 contains 101 object classes and 1 background class, with 2416 samples in the train set and 6728 samples in the test set. The number of samples per class range from 17 to 30, and thus this is a suitable dataset to highlight the practical importance of adversarial self-supervised pretrained representations for low-data regime.
> For Adversarial Full Finetuning (AFF), a base robustly pretrained model is finetuned using the TRADES adversarial training for 25 epochs. We present results on WideResNet-34-10 models that are pretrained on CIFAR-10 and CIFAR-100 respectively. (SA: Standard Accuracy, RA-PGD20: Robust Accuracy against PGD-20, RA-G: Robust Accuracy against GAMA)
>
>
> |                     | | CIFAR-10 -> STL10          |       |  |  CIFAR-100 -> STL10        |       |
> |---------------------|:-----------------:|:--------:|:-----:|:------------------:|:--------:|:-----:|
> |                     |         **SA**        | **RA-PGD20** |  **RA-G** |         **SA**         | **RA-PGD20** |  **RA-G** |
> | Supervised (TRADES) |       64.58       |   39.83  | 32.78 |        64.22       |   34.20  | 31.01 |
> |        DeACL        | 61.65 | 31.88 | 28.34 | 60.89 | 33.06 | 30.00 |
> |       ProFeAT (Ours)      |       74.12       |   40.15  | 36.04 |        68.77       |   35.35  | 31.23 |
>
>
> |                     |  | CIFAR-10 -> Caltech101         |       | |  CIFAR-100 -> Caltech101         |       |
> |---------------------|:----------------------:|:--------:|:-----:|:-----------------------:|:--------:|:-----:|
> |                     |           **SA**           | **RA-PGD20** |  **RA-G** |            **SA**           | **RA-PGD20** |  **RA-G** |
> | Supervised (TRADES) |          62.46         |   40.77  | 39.40 |          64.97          |   42.95  | 41.02 |
> |        DeACL        |          62.65         |   41.39  | 39.18 |          61.01          |   40.56  | 39.09 |
> |       ProFeAT (Ours)      |          66.11         |   45.29  | 42.12 |          64.16          |   42.95  | 41.25 |
>
>  - We note that the proposed method outperforms DeACL by a large margin. Further, we note that by using merely 25 epochs of adversarial finetuning using TRADES, the proposed method achieves improvements of around 4% on CIFAR-10 and 11% on CIFAR-100 when compared to the linear probing accuracy presented in Table-5, highlighting the practical utility of the proposed method. The AFF performance of the proposed approach is better than that of a supervised TRADES pretrained model as well.
>
> We sincerely hope our rebuttal addresses the reviewers concerns. We will be happy to answer any further questions as well.

---

### Official Review · Reviewer_q8j3 · 2023-11-09

**Soundness:** 3 good
**Presentation:** 3 good
**Contribution:** 2 fair
**Rating:** 6
**Confidence:** 4

**Summary:**

This paper analyzed and improved the robustness of representation learned by self-supervised contrastive learning methods against adversarial attacks. They find a significant performance gap in the more prominent models (e.g., WideResNet-34-10) for the existing techniques. They attribute this to the mismatch between the objectives of the teacher and student models. And they find that an additional projector layer with some losses can help to mitigate such discrepancies and improve performance. The evaluations have been done on CIFAR-10 and CIFAR-100.

**Strengths:**

- The method is simple with the use of a projector network for adversarial training
- The effectiveness in the metrics standard accuracy (SA) and robustness against AutoAttack (AA) is encouraging

**Weaknesses:**

•	The novelty is marginal: The projector is widely used in SSL which can significant boost performance when evaluating the representations after the backbone network. Here, the use of such projector can improve the representation (as demonstrated in SA metric) is not surprising in self-supervised learning for adversarial attack. Also, the use of weak augmentation for the teacher and strong one for student is exploited in several prior works.
•	The robust accuracy (RA) is an important metric in evaluation of the robustness but it seems to be omitted in the main paper (table 3,4,5,6), could the author provide the results and analysis of this metric side by side, too?
•	The PGD-20 metric of the proposed method is pretty worse than the other SOTA in most cases but it is not adequately discussed or mentioned. Could the authors provide some intuitions why does such degradation on PGD-20 happen, any investigation to address that drawback?
•	Clarity: 1) It should be consistent in the style, for example, table 3, 5 where the bold results show the best performance but table 4, 6, … are not highlighted, making it hard to follow which one is better. 2) It should be also consistent to report the metric in table 3,4,5,6, for example, table 3 used the mixture “SA” and “AutoAttack”, while table 4 used the full metric “Standard Accuracy”, etc… the consistency should be done for all tables. 3) Since AA has been used for “AutoAttack”, it should be used differently for AA when referring the AutoAugment (table 8) to avoid confusing.

**Questions:**

See weakness

---

> ### Author Response · Authors · 2023-11-14
> **Response to reviewer q8j3 [1/2]**
>
> We thank the reviewer for their valuable feedback. We clarify their concerns below and upload a new version of the paper incorporating their feedback.
>
> -  Novelty:
>    - The removal of last few layers for better generalization was first introduced in a transfer learning setting in 2014 [1] and later adapted to self-supervised learning in the seminal work SimCLR [2]. Despite this, there have been several recent works on the use of projection layer and understanding its role in self-supervised and supervised learning, as highlighted by Bordes et al. (2023)[3], and this is still an active area of research. We propose to use a frozen pretrained projector for self-supervised adversarial training using distillation from a standard SSL trained network, which is significantly different from all existing works. We further hypothesize why the projector helps, and verify the same using well-designed experiments in both standard and adversarial self-supervised distillation.
>    - We find that the use of strong augmentations results in more diverse attacks as shown in Fig.2. While this results in better robust generalization, it also leads to a more complex training objective. The proposed approach maps the representations corresponding to these diverse strong augmentations at the student to the representations corresponding to weak augmentations (which are close to the unaugmented images) at the teacher, allowing the student to be invariant to strong augmentations without increasing the training complexity. Although a combination of weak and strong augmentations is used in other areas such as semi-supervised learning as well, the justification for using this strategy is different.
>
>     The effective use of augmentations in supervised and self-supervised *adversarial training* has been very challenging, and it is common to use the basic pad+crop augmentation alone for adversarial training [4-7]. Thus, although the use of weak and strong augmentations together is not new, we believe there is significant value in leveraging this for self-supervised adversarial training using a distillation framework, where the need for such a method is different from prior works.
>
>  - We list our contributions below for better clarity -
>
>     - Our finding that the use of a projection layer during self-supervised distillation can bridge the gap in clean accuracy between self-supervised adversarial training and supervised adversarial training methods
>     - Our hypotheses on why the projection layer helps, which is not only a valuable contribution to adversarial self-supervised learning but can also motivate a better understanding of the role of the projector in standard self-supervised distillation. We support our hypotheses using well-designed experiments in Tables-1 and 2.
>     - The use of weak and strong augmentations for the teacher and student respectively, to strike a balance between improving perturbation diversity and limiting training complexity.
>     - Understanding the role of different training losses at the feature and projector.
>     - Strong empirical results that push the state-of-the-art in self-supervised adversarial training by a large margin, and bridge the gap with respect to supervised adversarial training methods.
>
> [1] Jason Yosinski, Jeff Clune, Yoshua Bengio, and Hod Lipson. How transferable are features in deep neural networks? In Z. Ghahramani, M. Welling, C. Cortes, N. Lawrence, and K.Q. Weinberger (eds.), Advances
> in Neural Information Processing Systems, volume 27. Curran Associates, Inc., 2014.
>
> [2] Ting Chen, Simon Kornblith, Mohammad Norouzi, and Geoffrey Hinton. A simple framework for contrastive learning of visual representations. ICML 2020
>
> [3] Florian Bordes, Randall Balestriero, Quentin Garrido, Adrien Bardes, and Pascal Vincent. Guillotine regularization: Improving deep networks generalization by removing their head., TMLR 2023.
>
> [4] L. Rice, E. Wong, and J. Z. Kolter. Overfitting in adversarially robust deep learning. ICML 2020
>
> [5] S. Gowal, C. Qin, J. Uesato, T. Mann, and P. Kohli. Uncovering the limits of adversarial training against norm-bounded adversarial examples. arXiv preprint arXiv:2010.03593, 2020.
>
> [6] H. Zhang, Y. Yu, J. Jiao, E. Xing, L. El Ghaoui, and M. I. Jordan. Theoretically principled trade-off between robustness and accuracy. ICML 2019
>
> [7] Chaoning Zhang, Kang Zhang, Chenshuang Zhang, Axi Niu, Jiu Feng, Chang D Yoo, and In So Kweon. Decoupled adversarial contrastive learning for self-supervised adversarial robustness. ECCV 2022
>
>            (response continued in the next comment)

---

> > ### Author Response · Authors · 2023-11-14
> > **Response to reviewer q8j3 [2/2]**
> >
> > - Robust Accuracy metric: To measure robust accuracy, we present results against several well-known adversarial attacks. The widely accepted benchmark for measuring robustness is the AutoAttack, which is an ensemble of several strong white-box and black-box attacks. We report results against AutoAttack in the main SOTA comparison tables (Tables-3, 4, 5). However, evaluating a model against AutoAttack is computationally intensive. Thus, for the ablation experiments (Tables-6, 7, 8), we evaluate against GAMA, which is a single 100-step attack that is known to be competent with AutoAttack while being significantly faster. We clarify this in Section-2 of the updated draft and update the terminology used for robust accuracy in the tables as well.
> >
> > - PGD-20 accuracy: As discussed above, AutoAttack provides the best estimate of robustness, and GAMA (100 steps) closely matches it while being much faster. However, PGD-20 is a 20-step PGD attack, which is sub-optimal and does not give a correct robustness estimate. The gap between the true robustness of a model (which can be approximated by accuracy against AutoAttack or GAMA) and the robustness against an attack such as PGD-20 (with a small number of optimization steps) depends on the local linearity of the loss surface. For a perfectly linear loss surface, a single-step attack such as FGSM can also find the adversary effectively. However, for more convoluted loss surfaces, multiple attack steps with adaptively reducing step size and better loss functions are required to find the true loss maxima. Thus, a lower gap between PGD-20 and AutoAttack implies better smoothness of the loss surface. Therefore, when the proposed method has better robust accuracy against AutoAttack and worse accuracy against PGD-20 (For eg. DeACL vs. Ours in Table-3), the proposed method has a smoother loss surface (due to better robustness) and thus PGD 20 step attack is able to reliably find the adversarial attack that exists. Thus, this is not a drawback, and is in fact a demonstration of better robustness of the model. We clarify this in Section-2 of the updated draft.
> >
> > - Consistency in tables: In the updated draft, we bold the best results and improve consistency in naming the columns as suggested.
> >
> > We would be happy to answer any further questions as well.

---

> > > ### Author Response · Authors · 2023-11-23
> > > **A gentle reminder to reviewer q8j3**
> > >
> > > We thank the reviewer for appreciating the simplicity and effectiveness of the proposed approach, and the presentation of our paper. We post a gentle reminder to the reviewer to review our response and hope that our rebuttal addresses their concerns. We will be happy to answer any further questions as well.

---

### Author Response · Authors · 2023-11-14
**A note to all reviewers**

We sincerely thank the reviewers for their time and valuable feedback on our work. We are happy to see that the reviewers find our work novel [e5nc], simple [q8j3] and effective [q8j3, s59e, e5nc, eRyw], well-written [q8j3, s59e, eRyw], our evaluations and ablations to be insightful and thorough [s59e, e5nc, eRyw], and our results to be strong and convincing [s59e, eRyw]. We have uploaded an updated draft to incorporate the reviewers' suggestions, with changes highlighted in blue. We plan to post our responses one by one (as we finish writing them), in order to utilize the author-reviewer discussion window effectively.

---

> ### Author Response · Authors · 2023-11-21
> **A gentle reminder to review our rebuttal**
>
> Dear Reviewers,
>
> We sincerely thank you for your time and valuable feedback on our work. We believe that we have addressed all the concerns/ questions in the rebuttal and eagerly look forward to hearing your thoughts on the same. We have also updated our draft based on the feedback received. We request you to kindly review our responses and update the scores if our rebuttal addresses your concerns.
>
> Thanks!

---

### Author Response · Authors · 2023-11-22
**Request to review our rebuttal**

We thank the reviewers once again for their efforts in reviewing our work. We have responded to their concerns and also updated the draft based on their feedback.

Since we are close to the end of the author-reviewer discussion period, we sincerely request the reviewers to share their feedback on our rebuttal, which will give us an opportunity to respond to their feedback.

Thanks!

---

### Author Response · Authors · 2023-11-23
**A gentle reminder to update scores**

We hope our rebuttal has addressed the reviewers concerns. We understand that the ICLR discussion timing this year has been very tight due to which we could not get time for discussion on the paper. Nevertheless, we request the reviewers to kindly review our rebuttal and update their scores if it answers their concerns. We will try our best to respond to any  further questions that come up in the next few hours.

---

### Meta-Review · Area_Chair_Bmcg · 2023-12-07

**Metareview:**

This paper received mixed reviews of 5, 5, 6, 6. The major concerns raised by the reviewers were:

1. marginal novelty - projector is widely used in SSL, weak/strong augmentation for teacher/student also used before. [q8j3]
2. robust accuracy is not included in the main table. [q8j3]
3. why is the PGD-20 metric worse for the proposed work compared to SOTA? [q8j3]
4. various presentation issues [q8j3]
5. ablation study does not show the specific contribution of each component. [s59e]
6. more interesting to see results on low-data tasks, such as Caltech [s59e]
7. further comparison needed with other evaluation techniques (KNN), and the effectiveness of pre-training on downstream tasks with fine-tuning [e5nc]
8. Section 4.1 lacks compelling evidence and in-depth analysis [e5nc]
9. lacks robust analysis [e5nc]
10. methodology lacks strong justification for why this combination is most effective [e5nc]
11. why the method is more effective for larger models? [e5nc, eRyw]
12. unfair comparison with DynACL (different number of epochs) [eRyw]

The authors wrote a response to address these concerns. During the discussion, the negative reviewers were not assuaged by the response. Specifically, Reviewer e5nc was not convinced by the justification/motivation of the proposed approach, and found the authors response about distillation to be generic, and how aligning linear probing is advantageous for robustness is unclear.  Reviewer e5nc also disagreed about defining a specific downstream task, and noted that the augmentation actually was the key component rather than the projector (the ablation results in response to Rev s59e's question), which mismatches the paper claims. Furthermore, two reviewers found the work was incrementally novel, combining various components together, which are used already in SSL. However, after the response, one Rev q8j3 was satisfied and raised their rating to 6.

Overall, the AC agrees with the concerns of the reviewers, and thus recommends reject.

**Justification For Why Not Higher Score:**

- the novelty is incremental -- reusing projectors and weak/strong data augmentation for this specific task.
- justification/motivation of the approach is not convincing enough.

**Justification For Why Not Lower Score:**

n/a

---

### Decision · Program_Chairs · 2024-01-16

Reject